# MODEL COLLAPSE IS NOT A BUG BUT A FEATURE IN MACHINE UNLEARNING FOR LLMS

**Yan Scholten**[1]**, Sophie Xhonneux**[2]**, Leo Schwinn**[*, 1]**, Stephan Günnemann**[*, 1]
[1]Dept. of Computer Science & Munich Data Science Institute, Technical University of Munich
[2]Mila, Université de Montréal
`{y.scholten, l.schwinn, s.guennemann}@tum.de, lpxhonneux@gmail.com`

## ABSTRACT

Current unlearning methods for LLMs optimize on the private information they seek to remove by incorporating it into their fine-tuning data. We argue this not only risks reinforcing exposure to sensitive data, but also fundamentally contradicts the principle of minimizing its use. As a remedy, we propose a novel unlearning method—Partial Model Collapse (PMC), which does not require unlearning targets in the unlearning objective. Our approach is inspired by recent observations that training generative models on their own generations leads to distribution collapse, effectively removing information from model outputs. Our central insight is that model collapse can be leveraged for machine unlearning by deliberately triggering it for data we aim to remove. We theoretically analyze that our approach converges to the desired outcome, i.e. the model unlearns the data targeted for removal. We empirically demonstrate that PMC overcomes four key limitations of existing unlearning methods that explicitly optimize on unlearning targets, and more effectively removes private information from model outputs while preserving general model utility. Overall, our contributions represent an important step toward more comprehensive unlearning that better aligns with real-world privacy constraints.[1]

## 1 INTRODUCTION

Privacy regulations and copyright laws (e.g. the GDPR (European Union, 2016)) necessitate the ability to selectively remove data from machine learning models, including Large Language Models (LLMs). While complete retraining without specific data can be optimal for information removal, it is infeasible at scale given the high computational costs of training LLMs. This motivates the need for machine unlearning techniques to erase specific information while preserving a model's broader capabilities.

Although recent methods have demonstrated early progress in LLM unlearning—either via refusal fine-tuning or gradient ascent on ground-truth sequences (Zhang et al., 2024)—they degrade model utility and lack deeper theoretical analysis and robustness (Liu et al., 2025). In particular, we argue that optimizing on ground-truth sequences to be unlearned is counterintuitive and contradicts the principle of minimizing the use of private data. Critically, we show that this dependency can introduce side effects that remain poorly understood, such as enabling adversaries to elicit data after unlearning. These limitations highlight the need for novel unlearning methods that mitigate such risks.

In this paper, we identify notable parallels between the unlearning challenge and the phenomenon known as *model collapse*, where iterative fine-tuning on synthetic data causes information loss in the model's output distribution and can lead to distribution collapse (Shumailov et al., 2023; 2024; Bertrand et al., 2024; Ferbach et al., 2024). We raise the following critical research question:

*Can we leverage the principles underlying model collapse to develop*
*principled approaches for machine unlearning?*

To address this research question, we introduce **Partial Model Collapse (PMC)**—a fundamentally novel approach to machine unlearning that leverages the principles of model collapse. By iteratively fine-tuning the model on its own generations in response to sensitive questions, we can force the model's distribution to collapse on private data in a targeted manner, thereby unlearning it (Figure 1).

---

[*]Equal supervision
[1]Project page: `https://www.cs.cit.tum.de/daml/partial-model-collapse/`

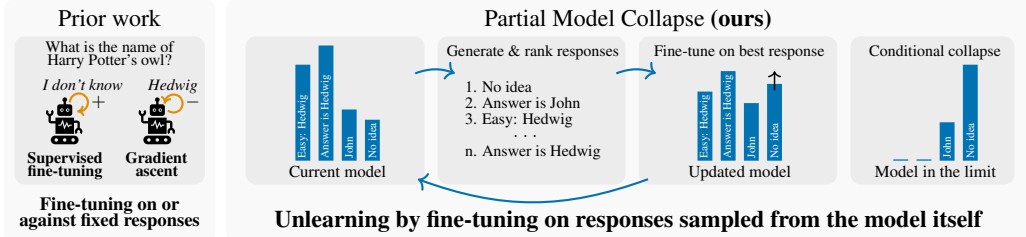

Figure 1: We propose Partial Model Collapse (PMC), a novel unlearning method that leverages the principles of model collapse to remove information from LLMs. By **iteratively fine-tuning LLMs on their own generated responses**, we trigger distribution collapse conditionally for sensitive questions, effectively removing information from model outputs. Unlike (1) fine-tuning on fixed refusals such as "I don't know", or (2) using gradient ascent to optimize against fixed ground-truth sequences, PMC fine-tunes on responses the model is already likely to generate. This allows us to achieve more effective and robust unlearning without requiring fixed ground-truth sequences in the fine-tuning data.

We provide theoretical analysis showing that our approach achieves unlearning by converging to the desired outcome. We begin by motivating the method on categorical data, then extend it to arbitrary distributions, and ultimately adapt it for practical use in LLMs for question-answering tasks.

In extensive experimental evaluations we demonstrate that PMC removes information from model outputs more effectively than existing methods while being theoretically principled. Notably, PMC overcomes four key limitations of prior approaches: *First*, it is more robust against sampling and prefilling attacks. *Second*, it preserves generation coherence by avoiding unintended degradation in unrelated contexts. *Third*, it reduces information leakage by preventing unnatural suppression of correct answers, thereby mitigating vulnerability to probability-based attacks. *Fourth*, it reduces information leakage in the presence of sampling and prefilling attacks. Our main contributions are:

- We propose **Partial Model Collapse (PMC)**—a novel, theoretically grounded unlearning method based on **iterative relearning on synthetically generated data**.

- We provide a **formal analysis showing that PMC achieves unlearning** by driving the model's output distribution toward a target distribution in which the influence of private data is eliminated.

- We identify **negative side effects in previous, target-dependent unlearning methods**, including distorted token probabilities for unlearning targets even out of context of the unlearning task and information leakage regarding supposedly unlearned knowledge in multiple choice evaluations.

- Through extensive empirical evaluation, we show that **PMC outperforms existing state-of-the-art unlearning methods** in removing information from LLM outputs. It maintains generation coherence across tasks and shows no negative side effects that we identify in previous methods.

Overall, we introduce a new paradigm for machine unlearning by harnessing the mechanism of model collapse. By reframing this detrimental phenomenon as a tool for targeted information removal, we enable new avenues toward more trustworthy machine learning.

## 2 RELATED WORK

**Machine unlearning.** Broadly, machine unlearning can be categorized into exact, approximate, and empirical methods. Exact unlearning seeks to ensure that the resulting model behaves as if specific data had never been seen (Bourtoule et al., 2021; Yan et al., 2022), but is typically computationally infeasible at scale. Approximate unlearning, while not guaranteeing complete removal, aims to reduce the influence of specific data points statistically, often drawing on tools from differential privacy (Guo et al., 2020; Neel et al., 2021; Sekhari et al., 2021; Ullah et al., 2021; Chien et al., 2022; Zhang et al., 2023). In contrast, empirical unlearning typically pursues more practical objectives. One common objective is to converge in data distribution toward a model that was never trained on specific data (Maini et al., 2024). Another critical objective (which we focus on) aims to remove specific information from model outputs (Eldan & Russinovich, 2023; Krishnan et al., 2025). The empirical nature of these methods makes them scalable to larger models including LLMs (Jang et al., 2022).

**Machine unlearning for LLMs.** Recent research has increasingly focused on unlearning in the context of LLMs (Jang et al., 2022; Chen & Yang, 2023; Eldan & Russinovich, 2023; Kim et al., 2024; Lynch et al., 2024; Sheshadri et al., 2024; Li et al., 2024; Seyitoğlu et al.; Shi et al., 2025; Dorna et al., 2025). Among empirical approaches, methods based on preference optimization have shown early progress (Rafailov et al., 2023; Zhang et al., 2024; Fan et al., 2024; Mekala et al., 2024), yet all of them introduce severe unlearning-utility trade-offs. Moreover, evaluating unlearning in LLMs remains an open challenge (Feng et al., 2025; Jones et al., 2025; Scholten et al., 2025). Most current methods focus on assessing the model's ability to avoid generating specific unlearning targets, but often overlook issues such as residual information leakage (Schwinn et al., 2024; Scholten et al., 2025). In this work, we identify further negative side effects in current methods.

**Model collapse in iterative retraining.** The rise of AI-generated content on the web has sparked growing interest in the effects of iterative retraining, where models are repeatedly trained on their own outputs. Early studies (Shumailov et al., 2023; Alemohammad et al., 2024) raised concerns by showing that model performance can degrade significantly with successive retraining iterations. In contrast, Bertrand et al. (2024) show that mixing synthetic data with the original training data can avoid model collapse and stabilize performance. Theoretical work (Dohmatob et al., 2024; Feng et al., 2024) further derives conditions under which collapse occurs. For example, iterative retraining with discrete or Gaussian distributions results in collapse primarily due to statistical approximation errors (Shumailov et al., 2023; Alemohammad et al., 2024; Bertrand et al., 2024). Most recently, Ferbach et al. (2024) introduce a new model for retraining in practice, where new synthetic training data is sampled according to a Bradley-Terry model with an unknown reward function. They show that retraining maximizes the underlying reward function and that mixing synthetic and original training data can prevent collapse. While model collapse has been framed as a bug in the LLM learning landscape, we show that it can be turned into a feature in the context of machine unlearning.

## 3 PRELIMINARIES AND BACKGROUND

**Machine unlearning.** In this work, we focus on empirical machine unlearning for LLMs, defining it as the problem of removing information from model outputs without retraining from scratch and, in contrast to previous works, without requiring access to ground-truth responses to sensitive questions.

**Large language models.** We model LLMs as parameterized functions $f_\theta : V^* \to \mathcal{P}(V^*)$ mapping input queries of arbitrary length to distributions over output sequences given vocabulary $V$, where $*$ is the Kleene operator. Output distributions can only be evaluated sequentially, i.e. the probability of output sequence $y = (y_1, \ldots, y_m)$ given input $x$ is the product of conditional next-token probabilities, $f_\theta(y|x) = \prod_{i=1}^m f_\theta(y_i|y_{i-1}, \ldots, y_1, x)$, where $f_\theta(y_i|\cdot)$ is the PMF over possible tokens $y_i \in V$.

**Iterative relearning on self-generated data.** Given an initial generative model $f^{(0)}$ fitted on a dataset $D^{(0)}$, iterative relearning refers to sequentially fine-tuning models on data sampled from their own distribution $\{x_i \mid x_i \sim f^{(t)}\}_{i=1}^n$ to produce models of the next generation $f^{(t+1)}$. The goal is to study the limit behavior of the sequence $f^{(1)}, f^{(2)}, \ldots, f^{(t)}$ for $t \to \infty$. In this context, model collapse refers to the phenomenon that iterative relearning causes loss of information over time, and eventually leads to model collapse (Shumailov et al., 2023; 2024), i.e. the variance of the model's generative output distribution vanishes in the limit, $\mathrm{Var}_{y \sim f^{(t)}}[y] \overset{t \to \infty}{\longrightarrow} 0$.[2]

**Discrete preference models.** Ferbach et al. (2024) study the stability of iterative relearning on curated self-generated data in the image domain. They model the curation process using a reward function and the Bradley-Terry model (Bradley & Terry, 1952), which is a probabilistic model for pairwise comparisons of items and often used to model human preferences. The model formulates the probability of one item $x_1$ being preferred over another $x_2$ using item-dependent scores (Bradley & Terry, 1952). Given $n$ items $x_i$, the probability of choosing $\hat{x} \sim \mathcal{BT}_\tau(x_1, \ldots, x_n)$ under the generalized Bradley-Terry model $\mathcal{BT}_\tau$ with temperature $\tau$ can be described as

$$\Pr_{\hat{x} \sim \mathcal{BT}_\tau(x_1, \ldots, x_n)}[\hat{x} = x_i] = \frac{e^{r(x_i)/\tau}}{\sum_{j=1}^n e^{r(x_j)/\tau}}, \tag{1}$$

where $r(x)$ is a reward function that assigns a score to each item $x_i$. Our approach uses this preference model to guide the unlearning process by choosing samples with higher unlearn quality.

---

[2]Note that we consider collapse of the model's output distribution, not of the model's overall utility.

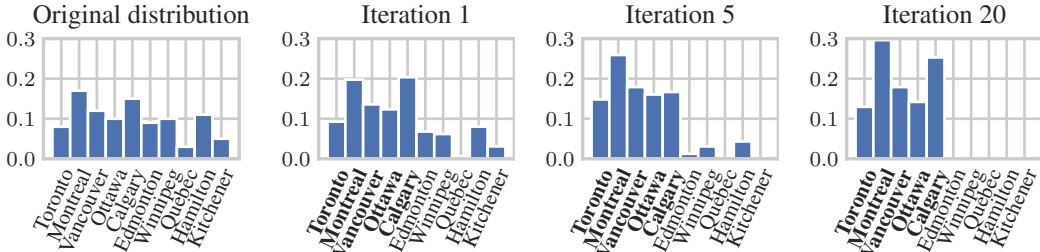

Figure 2: Unlearning through iterative MLE-relearning for categorical distributions. The model's knowledge about all other categories vanishes over time until it models target categories (bold) only.

## 4 FROM MODEL COLLAPSE TO MACHINE UNLEARNING

In the following, we theoretically motivate and derive a new perspective on machine unlearning that leverages information loss caused by iterative relearning on self-generated data.

### 4.1 WARM-UP: UNLEARNING IN CATEGORICAL DISTRIBUTIONS VIA ITERATIVE RELEARNING

We begin by analyzing iterative relearning of categorical distributions via maximum likelihood estimation (MLE). Assume a dataset $\mathcal{D}$ of categorical data with at least one datapoint per category, and an initial categorical distribution $\pi_0$ fitted on $\mathcal{D}$ using MLE. We further define a subset $\mathcal{D}_\mathcal{C} \subseteq \mathcal{D}$ of datapoints belonging to target categories $\mathcal{C}$ and delete all other datapoints from the dataset. We then introduce an iterative relearning process that fits a new categorical distribution $\pi_{t+1}$ on the target data $\mathcal{D}_\mathcal{C}$ augmented with self-generated data, i.e. datapoints generated from the distribution $\pi_t$ of the previous iteration: $\mathcal{D}_\mathcal{C} \cup \{x_i \mid x_i \sim \pi_t\}_{i=1}^n$. Interestingly, this iterative relearning prevents total distribution collapse, causes information loss for all other categories and effectively achieves full unlearning of the deleted datapoints (Proof in Appendix C):

> **Lemma 1:** *For any categorical distribution $\pi_0$, iteratively relearning $\pi_t$ on target data $\mathcal{D}_\mathcal{C}$ augmented with data generated from its own distribution $\{x_i \mid x_i \sim \pi_t\}_{i=1}^n$ causes information loss for all other (non-target) categories $i$: $\pi_t(i) \xrightarrow{t \to \infty} 0$.*

Intuitively, the probability mass of the other categories gets redistributed to the target categories and results in a "partial" collapse (Figure 2). Without target data, i.e. $\mathcal{D}_\mathcal{C} = \emptyset$, the iterative relearning process would converge to total distribution collapse (Shumailov et al., 2023; 2024), i.e. the model would eventually assign all probability mass to a single category. The main reason for this information loss are statistical approximation errors when fitting categorical distributions using maximum likelihood estimation: Given finite samples, the iterative relearning process describes an absorbing Markov chain, which is known to converge to an absorbing state (Shumailov et al., 2023; 2024).

### 4.2 MACHINE UNLEARNING VIA ITERATIVE RELEARNING ON SELF-GENERATED DATA

Our core idea is to leverage this inherent information loss described above for machine unlearning, gradually forcing the model to forget undesired responses without explicitly optimizing against ground-truth sequences. However, this comes with several challenges for LLMs in practice:

*First*, the distributions we seek to collapse for LLMs are the categorical distributions over entire sequences $\mathcal{P}(V^*)$, but LLMs only provide direct access to the categorical next-token distribution. *Second*, LLM unlearning is typically studied for question-answering tasks, where the objective is to unlearn answers to "forget" questions while preserving performance on all other "retain" queries. *Lastly*, defining a suitable target distribution to converge to is challenging due to the natural language domain—although we might know which answers should be unlearned, specifying a well-formed distribution to converge to remains non-trivial without access to a language model that has not been trained on the ground truth (which is usually not available without expensive retraining from scratch).

**Partial model collapse using preference optimization.** To overcome these challenges, we propose to trigger collapse of the model's output distribution conditional on forget queries through an iterative preference-guided procedure while ensuring that the model retains its utility on other retain queries.

To guide the unlearning process toward desired outputs, we build upon the result that iterative retraining on "curated" (filtered) self-generated data yields model collapse in the image domain (Ferbach et al., 2024). Specifically, we propose to unlearn responses to forget queries by (1) sampling $n$ independent responses from the model, and (2) fine-tuning on the best response selected by a preference model. We formalize this using the generalized Bradley-Terry preference model (Section 3) together with a bounded reward function $r : \mathcal{X} \to [0, r^*]$, which assigns higher scores to preferred responses (e.g. rewarding dissimilarity of a sampled response to the response of the original model).

Let $p_r$ represent a retain distribution over query-answer pairs (which we do not want to unlearn), and $p_f$ a forget distribution over questions whose answers we want to unlearn. Note that we do not require access to ground-truth answers to forget questions, and we assume disjoint support of $p_f(q)$ and the marginal distribution $p_r(q)$, i.e. we either want to unlearn the response to a question or not. Given an initial model $p_0$ before unlearning, we introduce the following iterative unlearning process:

---

**Partial Model Collapse Machine Unlearning for Q&A tasks**

$$p_{t+1} = \arg\max_{p \in \mathcal{P}} \; \lambda \mathbb{E}_{(q,x) \sim p_r}[\log p(x|q)] + \mathbb{E}_{\substack{q \sim p_f \\ x_1,\ldots,x_n \sim p_t(x|q) \\ \hat{x} \sim \mathcal{BT}_\tau(x_1,\ldots,x_n)}} [\log p(\hat{x}|q)] \quad (2)$$

---

where $\mathcal{P}$ is the set of all distributions over $\mathcal{X}$, $p_t$ is the model distribution at step $t$, and $\mathcal{BT}_\tau$ is the generalized Bradley-Terry preference model with temperature $\tau$ (Equation 1). Intuitively, Equation 2 describes an iterative unlearning process where the next distribution maximizes the expected log-likelihood of question-answer queries under the retain distribution $p_r$ (for utility) and the expected log-likelihood of curated samples from the current model distribution $p_t$ conditioned on forget queries from $p_f$ (for unlearning). The first term preserves utility and the second term is responsible for unlearning, where the parameter $\lambda \in [0, \infty)$ balances the trade-off between utility and unlearning. Notably, this iterative process defined in Equation 2 converges to the maximum reward for any forget query $q \in supp(p_f)$ in the limit, i.e. the model unlearns:

---

**Theorem 1:** *Let $p_t$ be the distribution described by Equation 2 and assume non-zero probability mass on the maximum reward $\Pr_{x \sim p_0(x|q)}[r(x) = r^*] > 0$ for forget queries $q \in supp(p_f)$. In the absence of statistical and function approximation errors, the expected reward converges to the maximum reward and its variance vanishes for any forget query $q \in supp(p_f)$:*

$$\mathbb{E}_{x \sim p_t(x|q)} \left[ e^{r(x)} \right] \xrightarrow{t \to \infty} e^{r^*} \qquad \mathrm{Var}_{x \sim p_t(x|q)} \left[ e^{r(x)} \right] \xrightarrow{t \to \infty} 0.$$

---

Intuitively, the expected reward increases each iteration (proof in Appendix D).

### 4.3 PARTIAL MODEL COLLAPSE UNLEARNING FOR LLMs IN PRACTICE

Finally, we describe our proposed PMC unlearning loss in Algorithm 1, which can be minimized using standard (stochastic) gradient-based fine-tuning methods. Note that while Equation 2 provides a novel theoretical perspective, in practice LLMs are parameterized functions $f_\theta$ approximating $p_t$, and $p_r$ and $p_f$ are approximated via finite-sample datasets, denoted as the retain set of Q&A pairs $D_r = \{(q_i, x_i)\}_{i=1}^{m_r}$ and the forget set $D_f = \{q_i\}_{i=1}^{m_f}$ of questions whose answers we aim to unlearn.

Importantly, our unlearning loss is independent of the ground-truth forget answers, thereby avoiding any direct gradient updates that could unintentionally reinforce the information we seek to remove. Instead, we fine-tune on answers generated by the model itself. Specifically, we sample $n$ responses from the model's output distribution and select one response according to a preference model. The key advantage of our approach is that the samples are drawn directly from the model's own distribution—they represent outputs the model is already likely to produce. As a result, fine-tuning on these samples aligns with the model's distribution. Rather than pushing the model away from specific targets, we allow it to diverge naturally by adjusting the likelihood of its own likely generations, enabling unlearning while preserving the model's utility.

---

**Algorithm 1** PMC unlearning loss

**Require:** Retain batch $\mathcal{B}_r = \{q_i, x_i\} \subseteq D_r$, forget batch $\mathcal{B}_f = \{q_i\} \subseteq D_f$, model $f_\theta$, temperature $\tau$, and hyperparameter $\lambda$
1: Compute retain loss $\ell_r$
   $\ell_r = -\frac{1}{|\mathcal{B}_r|} \sum_{(q_i,x_i) \in \mathcal{B}_r} \log f_\theta(x_i|q_i)$
2: **for** forget question $q_i \in \mathcal{B}_f$ **do**
3:    Sample $n$ responses
      $x_1, \ldots, x_n \sim f_\theta(x|q_i)$
4:    Sample preferred response
      $\hat{x}_i \sim \mathcal{BT}_\tau(x_1, \ldots, x_n)$
5: Compute forget loss $\ell_f$
   $\ell_f = -\frac{1}{|\mathcal{B}_f|} \sum_{q_i \in \mathcal{B}_f} \log f_\theta(\hat{x}_i|q_i)$
6: **return** $\lambda \ell_r + \ell_f$

---

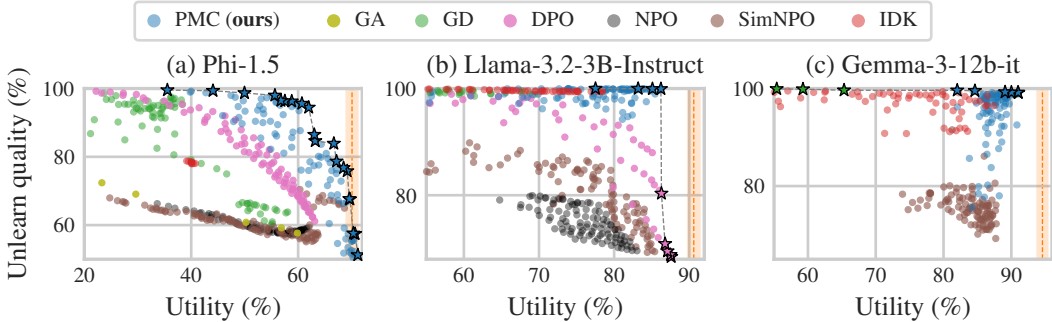

Figure 3: Partial model collapse (PMC) significantly dominates baselines and expands the Pareto-front w.r.t. utility and unlearn quality for (a) Phi-1.5, (b) Llama-3.2-3B-Instruct and (c) Gemma-3-12b-it. While existing methods (GA, GD, DPO, NPO, SimNPO, and IDK) also unlearn, they cannot deviate much from the fine-tuned model without compromising the model's general capabilities. Orange vertical lines indicate utility of fine-tuned models before unlearning. Stars represent dominating points. For improved accessibility we provide this plot with symbols instead of colors in Appendix A.

## 5 EXPERIMENTAL EVALUATION

We experimentally demonstrate that the information loss in model collapse can be leveraged to achieve machine unlearning for LLMs. We also identify negative side effects in existing unlearning methods that directly optimize on ground-truth sequences and showcase positive effects of our approach, such as robustness and reduced leakage under sampling. We provide additional results in Appendix A, and refer to Appendix B for experimental setups, implementation details and reproducibility instructions.

**Experimental setup.** We use the TOFU dataset (Maini et al., 2024), a fictitious dataset of 4,000 question-answering pairs designed for machine unlearning. We fine-tune models on the full dataset and perform unlearning on the "forget10" split, since it has the largest forget set and thus corresponds to the most challenging split in the dataset. We provide results for an additional dataset in Appendix A.

**Models.** We perform experiments for the following three models: Phi-1.5 (Li et al., 2023) since it is a smaller and extensively studied model in the unlearning literature, and Llama-3.2-3B-Instruct (Grattafiori et al., 2024) as well as Gemma-3-12b-it (DeepMind, 2025) since they are more recent models with strong performance across tasks. We run experiments on A100, H100 and H200 GPUs.

**Baselines.** As baselines we consider *Gradient Ascent* (GA), *Gradient Difference* (GD) (Liu et al., 2022), *Negative Preference Optimization* (NPO) (Zhang et al., 2024) and its simplified form (SimNPO) (Fan et al., 2024). We also compare to two baselines introduced in (Zhang et al., 2024): "*I don't know.*" (IDK), which fine-tunes on this fixed refusal response, and *Direct Preference Optimization* (DPO) (Rafailov et al., 2023), which uses IDK-phrases as positive and the ground truth as negative examples.

**Metrics**. We evaluate using recall ROUGE-L scores (Lin, 2004), i.e. the longest common subsequence between the model's greedy output and the ground truth. Unlearning performance is quantified using the sum of ROUGE-L scores on the forget and paraphrased-forget sets—the latter is an additional TOFU dataset allowing to quantify generalization of unlearning. We report *unlearn quality* as the maximal score minus the achieved score (such that larger is better), and *utility*, measured as the sum of ROUGE-L scores on the retain set $D_r$ and two additional TOFU datasets: world facts (117 questions) and real authors (100 questions), which allow to assess general knowledge retention. We approximate retain performance on $D_r$ using a (random but fixed) subset of 400 retain samples. We further normalize scores for better readability by dividing by the maximum possible score.

**Reward function.** The design of the reward function $r(x)$ can range from simple ROUGE-based rewards to more complex reward models trained on human preferences and is highly application-dependent since it determines post-unlearning behavior. Our goal is to demonstrate the effectiveness of PMC in removing information from model outputs and we therefore choose ROUGE-based rewards for their simplicity. Specifically, we use the ROUGE-L score between the model's original and current output, i.e., $r(x) = 1 - \text{ROUGE-L}(x, y) \in [0, 1]$, where $y$ is the model's original (greedy-decoding) answer for forget question $q \in \mathcal{D}_f$ and $x$ is the sampled output as described in Algorithm 1.

## 5.1 PARTIAL MODEL COLLAPSE ACHIEVES MORE EFFECTIVE UNLEARNING

In a series of experiments we compare our proposed partial model collapse (PMC) to the baselines (GA, GD, DPO, NPO, SimNPO, and IDK). Since all methods involve multiple hyperparameters, we perform a grid search for all methods. To ensure a fair comparison, we explore 100 different configurations for each method, covering a broad range of hyperparameter combinations while keeping the number of trials consistent across methods (details in Appendix B). We repeat each experiment five times using different random seeds, and report mean utility and unlearn quality.[3]

Notably, PMC significantly dominates all baselines in the utility-unlearning trade-off and expands the Pareto-front, achieving strong unlearn quality while maintaining high utility across models (Figure 3). In contrast, previous methods achieve lower unlearn quality and/or compromise the model's utility. The strong performance of our method stems from its distribution-aware optimization strategy: Unlike the IDK-baseline, PMC fine-tunes on responses that are already likely under the model's own distribution. In contrast to gradient-ascent-based baselines (which repeatedly optimize against fixed ground-truth sequences), PMC fine-tunes on newly sampled sequences in each iteration and relies on the model collapse phenomenon to force the model's responses to diverge towards more desired ones.

We observe that PMC-unlearning frequently converges toward response patterns that fall into three broad categories across models: (i) hallucinations, (ii) gibberish, or (iii) generic refusals that indicate the absence of knowledge. Examples of the latter include "The answer is not available", "There is no public information" and "Specific details are not available" (despite the reward function not explicitly incentivizing such responses). Note that the objective of this paper is to demonstrate the effectiveness of PMC in removing information from model outputs, and we do not explicitly optimize for generation coherence or refusal patterns after unlearning. Future work can design reward functions that explicitly incentivize generic refusals, or one that penalizes hallucinations and gibberish.

## 5.2 PMC IS MORE ROBUST AGAINST SAMPLING AND PREFILLING ATTACKS

Notably, we demonstrate that PMC exhibits substantially greater robustness against sampling and prefilling attacks compared to prior approaches. To evaluate robustness under sampling, we draw 100 answers from the output distribution of the unlearned model, and compute the ROUGE-L score between each sampled and ground-truth response. We then compute the maximum (worst-case) ROUGE-L score per question and report the average across all forget questions (see Appendix B.4 for the full experimental setup). The results in Figure 4 show that PMC significantly reduces leakage under sampling, in stark contrast to existing methods. While the simple supervised fine-tuning IDK baseline also reduces leakage under sampling, this effect is largely superficial. To demonstrate this, we perform prefilling attacks in which the model is prompted with a forget question and forced to continue from the prefix "The answer is:". This

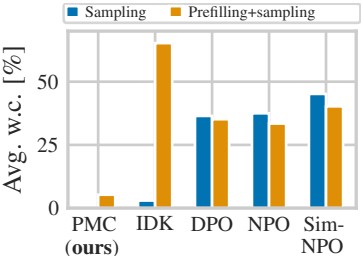

Figure 4: PMC is more robust against sampling and prefilling attacks. Lower average worst-case leakage is better.

attack bypasses the fine-tuned response and reveals that the IDK baseline still encodes substantial information about the unlearned answers, leading to high leakage. Notably, while existing methods can exhibit considerable leakage (Scholten et al., 2025), PMC is the first approach to achieve more robust unlearning across both attack settings. The underlying reason for the improved robustness is that PMC does not optimize on fixed sequences, but instead fine-tunes on newly sampled responses in each epoch. This allows PMC to achieve unlearning by leveraging the model collapse phenomenon, which leads to a more thorough divergence of the model's output distribution.

## 5.3 PMC OVERCOMES LIMITATIONS OF METHODS OPTIMIZING ON UNLEARNING TARGETS

Existing unlearning methods predominantly incorporate the unlearning target directly into their objectives. We argue that this approach may have subtle effects on model properties related to the unlearning targets, such as distorting token probabilities and leaking information about the private

---

[3]For Gemma-3-12b-it we run each experiment only once due to the high fine-tuning costs and do not report results for DPO/NPO as their scalability is limited due to their dependence on a reference model.

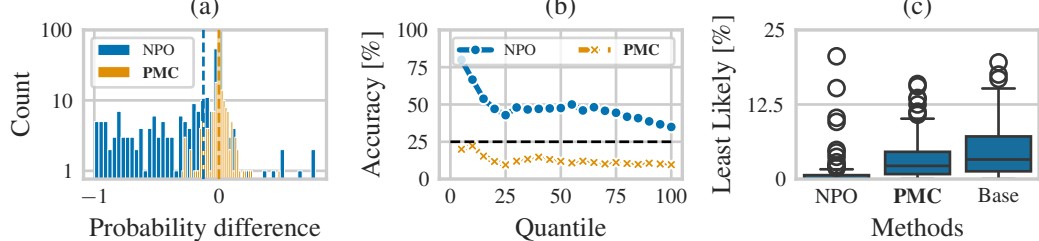

Figure 5: Limitations of unlearning methods optimizing on unlearning targets: (a) Side effects on unrelated datasets. (b) Accuracy when selecting least likely answer across quantiles (black line is random guessing). (c) Distribution of minimum probabilities across all multiple-choice options.

data used during unlearning optimization. Yet, the utility of unlearning models is typically evaluated using benchmark datasets or by comparing them to a retrained model (Maini et al., 2024). As a result, existing evaluations may miss subtle changes in the generation properties of unlearned models. In the following, we examine side effects of existing unlearning algorithms.

**Generation capability on unrelated datasets.** First, we study generations of tokens targeted in the unlearning optimization and investigate whether existing methods compromise the model's ability to generate such tokens. We argue that unlearning should prevent models from revealing unlearned information, but this effect must be limited to the unlearning context. It should not affect token generation in unrelated settings, as most tokens in forget sets are not semantically tied to the unlearning task but rather to sentence structure. For example, if we want to unlearn that John Doe is a carpenter, existing methods would minimize the probability of "carpenter" when asked about John Doe's profession. However, these methods should not reduce this probability in unrelated contexts.

To investigate such potential side effects, we compare the probability of generating tokens present in TOFU compared to the first 100 text chunks of the wikitext-2-raw-v1 train split (Merity et al., 2016). Figure 5 (a) shows the probability difference between unlearned models (NPO and our proposed PMC method) and the base model: $p_{un}(x_t|x) - p_{base}(x_t|x)$, where $x_t$ is a token present in the forget set, $x$ is the context of this token in the wikitext dataset, and $p(x_t|x)$ it the probability of $x_t$ given the context. As the base model, we use a model fine-tuned exclusively on the retain set, with no exposure to the forget data. NPO substantially reduces the probability of generating forget set tokens also present in wikitext. A considerable number of tokens that originally get assigned a high probability from the base model (e.g., close to 1) get assigned a probability of 0 from the unlearned model (indicated by $-1$ values in the figure). In contrast, our method preserves generation probabilities, exhibiting token probabilities that are neither systematically increased nor decreased. For PMC, the differences follow a zero-mean Gaussian distribution with small variance, whereas they are skewed to the left for NPO ($-0.12$ mean). This shows that methods dependent on unlearning targets can considerably distort token probabilities even out-of-context of the unlearning task.

**Probability distribution in multiple-choice settings.** Second, we hypothesize that existing unlearning methods may exhibit information "leakage" by unnaturally reducing the probability of correct answers, potentially allowing adversaries to identify forgotten information by simply selecting the least likely option. To further investigate such potential negative side effects, we created a multiple-choice dataset from the TOFU forget10 set by converting a subset of 84 questions into multiple-choice (MPC) format (Appendix B.5). We use the inverse perplexity of every answer as its score and turn scores into probabilities by normalizing them. Moreover, for the correct answers in the MPCs we used rephrased versions of the correct TOFU answers rather than exact matches to demonstrate that leakage can occur even for semantically similar but non-identical formulations.

Our experiments provide first empirical evidence for our leakage hypothesis. Figure 5 (b) shows accuracy when selecting the least likely answer across quantiles ordered by minimum probability among choices. NPO exhibits high accuracy for questions where the minimum probability is very low, indicating that the correct answer frequently becomes the least likely option. Conversely, our method shows no such pattern. Figure 5 (c) shows the distribution of minimum probabilities across all multiple-choice options. Here, NPO's distribution clusters near zero, further confirming that target-based unlearning unnaturally suppresses correct answer probabilities even in rephrased contexts.

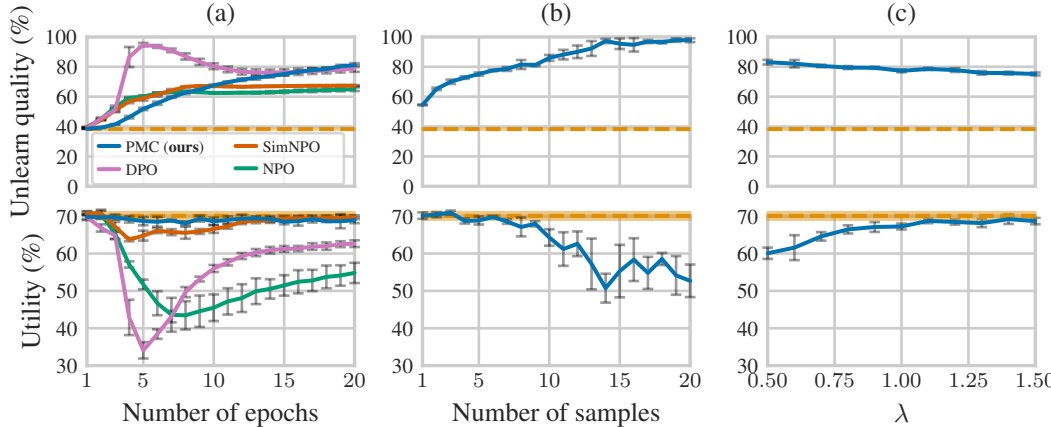

Figure 6: Ablation studies on (a) number of epochs, (b) number of samples, and (c) trade-off parameter $\lambda$. Dashed line is the fine-tuned model before unlearning. Shadows/bars indicate standard deviation.

### 5.4 EXTENDED EXPERIMENTAL EVALUATION OF COLLAPSE-BASED UNLEARNING

**Ablation studies.** We perform ablation studies on PMC's hyperparameters under Phi-1.5 (additional results in Appendix A, see also details in Appendix B). First, the number of training epochs strongly influences unlearning performance: while baseline methods converge after 10 epochs, PMC continues to improve unlearn quality without significantly affecting utility even after 20 epochs (Figure 6a). Second, increasing the number of samples enhances unlearning, with utility remaining stable for the first six epochs; larger sample sizes show higher variance in utility (Figure 6b). Finally, we ablate the unlearn-utility trade-off parameter $\lambda$, observing that larger values improve utility but can degrade unlearn quality, highlighting the importance of selecting $\lambda$ to balance these objectives (Figure 6c).

**Additional results.** We provide additional experiments in Appendix A: We provide further ablations in Appendix A.1, Appendix A.3 and Appendix A.4. We perform careful runtime comparisons in Appendix A.2, and demonstrate empirical reward convergence in Appendix A.6. We also provide experiments using self-BLEU-based rewards in Appendix A.7. Finally, we evaluate PMC on MUSE-news data in Appendix A.10, extending the analysis to an additional dataset beyond Q&A tasks.

**Extended utility analysis.** Although PMC is optimized only on the retain data to preserve utility, we find that its impact on overall model utility beyond the TOFU utility dataset is minimal in practice. Results on the ARC-Challenge, ARC-Easy, and MMLU benchmarks (Appendix A.8) show that PMC-unlearning has minimal to negligible effect on general model utility.

## 6 DISCUSSION

**Definition of machine unlearning for LLMs.** A common challenge in the LLM safety literature is that existing works often attempt to solve problems such as unlearning or alignment as monolithic objectives, an approach that frequently fails to capture nuances (Scholten et al., 2025). This motivates decomposing complex objectives into simpler, more measurable ones (Schwinn et al., 2025). In this context, machine unlearning for LLMs refers to a broader problem encompassing multiple objectives. A frequent goal in unlearning is to converge in data distribution toward a model never trained on the data to be unlearned (Maini et al., 2024). In this paper, we focus on the more direct and measurable objective of removing information from model outputs, which can contain undesired information even if models have never been trained on such data (e.g., through inference from other data). While our objective is closer to that of LLM alignment, it is a critical part of LLM unlearning. In particular, we argue that as long as models are not robust and leak under sampling (Scholten et al., 2025), convergence in data distribution is not achieved even if current benchmark evaluations suggest progress toward this goal. Moreover, we argue it is not sufficient to evaluate such convergence based on a single reference model fine-tuned exclusively on the retain data (i.e. not on forget data), since this process is non-deterministic and can lead to different models with substantially different properties.

Interestingly, we observe considerable utility loss at the point of distribution collapse from which the models quickly recover afterwards (see Appendix A.2), indicating convergence toward a model which may not be linearly mode connected to the model before unlearning. However, whether our PMC-unlearned models are actually closer to a model never trained on the data we removed from model outputs is a fundamentally different and harder question for which the current literature lacks comprehensive evaluation protocols, preventing robust conclusions so far.

**Evaluating LLM unlearning.** Evaluating unlearning in LLMs remains a major challenge beyond the definition of unlearning itself. Our analysis is inherently empirical and provides no formal guarantees that the model has unlearned the forget data. Currently, one can bound information leakage with high probability for fixed inputs (Scholten et al., 2025), but this also does not imply unlearning in a broader sense. While more comprehensive empirical evaluations would ideally rely on human (or LLM-based) judgments, our goal is simply to demonstrate that PMC can remove specific outputs without degrading utility—an outcome we believe is sufficiently supported by our experiments. We view this as a first step, and e.g. robustness to relearning or adversarial attacks remains future work.

**Limitations of collapse-based unlearning.** In theory, collapse-based unlearning relies on the model assigning non-zero probability mass to higher-reward responses so that the output distribution can shift toward more desirable generations. If the distribution were already fully collapsed onto a single ground-truth sequence, the reward could not increase further and we would not observe unlearning. While this constitutes a limitation in principle, we do not observe this in practice for LLMs. Empirically, current models maintain sufficiently broad output distributions with non-zero mass on higher-reward alternatives, enabling consistent convergence toward the optimal reward.

**Limitations of fine-tuning on samples.** While a key strength of PMC is its reliance on samples already likely under the model's own distribution, this also increases computational costs in particular for larger models. We provide a careful and detailed cost analysis in Appendix A, comparing PMC against baselines under runtime vs. unlearning and utility trade-offs. While we acknowledge that PMC has a slightly higher computational cost due to the initial collapse process in the first epochs, we believe that the overall runtime remains competitive and in particular practical for real-world applications. Future work could explore faster sampling techniques, pruned proxy models, or speculative sampling toward more efficient collapse-based machine unlearning.

**Privacy considerations.** A central advantage of PMC is that it neither optimizes against nor requires access to ground-truth forget sequences during unlearning. This is particularly important in scenarios where the original data is unavailable, restricted, or cannot be shared due to privacy constraints. Instead, PMC operates solely on samples drawn from the model's own distribution, eliminating the need for ground-truth supervision during unlearning. While self-generated samples may initially encode sensitive information, PMC theoretically and empirically drives the model to rapidly diverge from such content; after this initial phase, optimization no longer exposes the model to sensitive data. In contrast, prior GA-based methods repeatedly optimize against a fixed ground-truth sequence throughout the entire unlearning process, risking amplification of private information. More broadly, our setting reflects practical deployments in which one may not know which specific training samples gave rise to an output requiring unlearning, or where no unique ground-truth forget sequence exists.

**Design of reward function.** The design of the reward function $r(x)$ is crucial for achieving the desired outcome after unlearning. We use ROUGE-L scores between the current sampled and the model's original greedy output, which amounts to an incentive to diverge from the model before unlearning. This choice is motivated by (1) the goal of removing model outputs, as well as the (2) empirical effectiveness and (3) simplicity of this reward. In practice, the design of $r(x)$ may need to be tailored more carefully to the needs of specific applications, e.g. by using a fast proxy model that broadly captures coherence. We believe designing different rewards is a promising avenue for future work.

## 7 CONCLUSION

We propose a novel and theoretically grounded paradigm for LLM unlearning that leverages the model collapse phenomenon to remove information from model outputs. Our approach iteratively fine-tunes LLMs on their own responses to sensitive questions until the model's output distribution collapses on sensitive responses, effectively unlearning them. We empirically demonstrate that our approach converges to a model that no longer generates sensitive information while preserving the model's utility. Our work represents an important contribution toward effective unlearning and provides a foundation for future research in collapse-based unlearning for generative models beyond LLMs.

## ACKNOWLEDGMENTS

The authors want to thank Marius Mosbach, Alicia Curth, Marcel Kollovieh and Lukas Gosch for their valuable feedback on the manuscript. This work has been funded by the DAAD program Konrad Zuse Schools of Excellence in Artificial Intelligence (sponsored by the Federal Ministry of Education and Research). Leo Schwinn and Yan Scholten gratefully acknowledge funding from Coefficient Giving for this work. This research was enabled in part by compute resources provided by Mila. The authors of this work take full responsibility for its content.

## ETHICS STATEMENT

Our work contributes to the field of machine unlearning, which is crucial for ensuring privacy and compliance with data protection regulations. By proposing a method that effectively removes sensitive information from LLM outputs, we aim to enhance the trustworthiness of AI systems. However, we acknowledge that unlearning could also be misused, for example to delete facts. We advocate for the responsible use of our method, emphasizing transparency and accountability in AI development.

## REPRODUCIBILITY STATEMENT

We ensure reproducibility by providing a detailed description of our experimental setup (including all hyperparameters) and additional reproducibility instructions in Appendix B. All datasets, models, and code used in our experiments are publicly available. Our implementation is publicly available at https://www.cs.cit.tum.de/daml/partial-model-collapse/.

## LLM USAGE STATEMENT

LLMs were only used to polish writing at sentence-level (spelling, grammar, wording).

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

APPENDIX OVERVIEW

In this appendix we provide additional results and details on experimental setups, and prove theoretical results as outlined in the following: In Appendix A we provide additional experiments and ablation studies. We carefully describe experimental setups in Appendix B. Finally, we provide theoretical results in Appendix C and Appendix D.

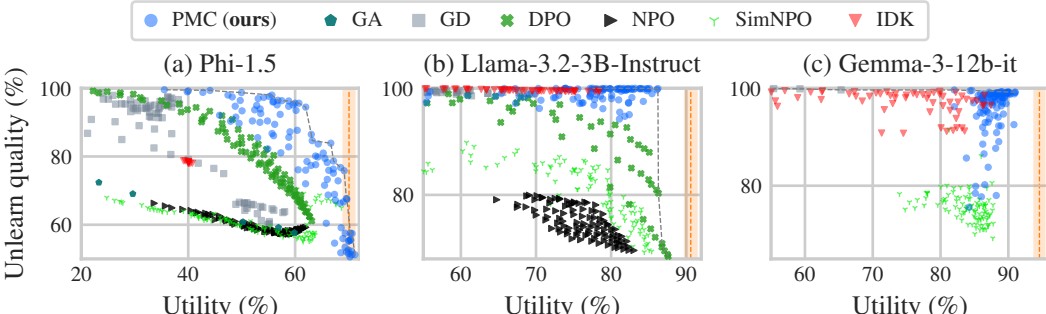

Figure 7: Partial model collapse (PMC) significantly dominates baselines and expands the Pareto-front w.r.t. utility and unlearn quality for (a) Phi-1.5, (b) Llama-3.2-3B-Instruct and (c) Gemma-3-12b-it. Same data as in Figure 3 but with symbols for improved accessibility.

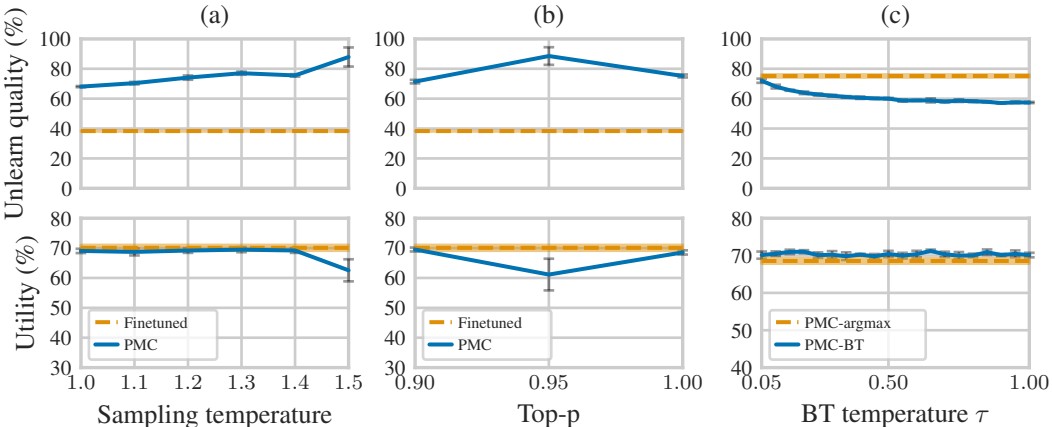

Figure 8: Ablation studies on: (a) temperature, (b) top-p sampling, and (c) Bradley-Terry approximation (shadow/bars show standard deviation across five runs).

## A    ADDITIONAL EXPERIMENTAL EVALUATIONS AND ABLATION STUDIES

In this section we provide additional experiments and ablation studies as follows:

- In Appendix A.1 we provide additional ablation studies on sampling temperature, top-p sampling and the Bradley-Terry approximation.
- In Appendix A.2 we provide a runtime analysis of PMC-unlearning compared to baselines.
- In Appendix A.3 we provide a detailed ablation study on the utility-unlearning trade-off parameter $\lambda$ and analyze unlearn quality and utility by epoch and runtime.
- In Appendix A.4 we provide a detailed ablation study on the number of samples and analyze unlearn quality and utility by epoch and runtime.
- In Appendix A.5 we empirically analyze the output distribution after unlearning of several unlearning methods to better understand robustness to sampling and prefilling attacks.
- In Appendix A.6 we analyze empirical reward convergence.
- In Appendix A.7 we experiment with an alternative reward function.
- In Appendix A.8 we provide an additional utility analysis of PMC-unlearned models.
- In Appendix A.9 we provide an ablation study on gradient checkpointing.
- In Appendix A.10 we extend the empirical analysis to MUSE-news data.

### A.1    CONTINUED ABLATION STUDY

We conduct the following additional ablation studies for Phi-1.5 to further investigate the properties of our method. See Appendix B.3 for details on the experimental setup of all ablation studies.

**Sampling temperature** (Figure 8 (a)). We empirically observe that larger temperatures allow stronger unlearn quality, likely due to the higher probability to sample responses with lower similarity to the ground truth. For temperatures above 1.5, the process leads to a decrease in utility.

**Top-p sampling** (Figure 8 (b)). We observe that top-p sampling can similarly improve unlearn quality at the cost of model utility due to similar effects on the diversity of the sampled responses.

**Bradley-Terry approximation.** The general PMC formulation in Section 4 requires randomly selecting one of $n$ responses according to the Bradley-Terry model. However, in all experiments we implement an approximation by choosing the response with the highest score, i.e., $\hat{x} = \arg\max_i r(x_i)$. We resolve ambiguity by choosing whichever sample has been drawn first to provide an additional, implicit incentive to choose more likely samples. Our approximation is computationally more efficient since it avoids the additional sampling step, and we empirically verify in Figure 8 (c) that it effectively corresponds to the limit $\tau \to 0$ for temperature $\tau$. In detail, the BT-temperature $\tau$ does not affect model utility, however, it effectively improves unlearn quality, motivating the argmax approximation.

## A.2 COMPUTATIONAL COST ANALYSIS

**Runtime vs. unlearning quality and utility.** We provide a computational cost analysis by visualizing runtime versus unlearning quality and utility plots for all three models in Figure 9, Figure 10 and Figure 11, respectively. Here we measure computational cost in terms of wall-clock runtime per epoch on an NVIDIA H200 GPU (all experiments in this section are performed on the same hardware). The plots show unlearning quality (left) and utility (right) versus runtime for different unlearning methods. Each point in the plots represents the end of an epoch during unlearning, and we connect the points of each method for better visualization. In total, we perform unlearning for 20 epochs for Phi-1.5 and Llama-3.2-3B-Instruct, and 10 epochs for Gemma-3-12b-it, which is more computationally expensive. `Phi-1.5` completes within 40 minutes, whereas `Llama-3.2-3B-Instruct` converges faster with a runtime of 20 minutes, and for `Gemma-3-12b-it`, the runtime is around 200 minutes. We consider the runtime of PMC to be practical for real-world applications, especially when compared to retraining LLMs from scratch. While for Phi-1.5 the runtime of PMC is higher than that of the baselines, for Llama-3.2-3B-Instruct and Gemma-3-12b-it, PMC is comparable to baselines (considering their runtime). We believe that the computational cost of PMC can be further improved by future research as discussed in Section 6, e.g. using more efficient sampling strategies.

**Regarding utility.** Note that all baselines affect utility during the unlearning process. In contrast to baselines, utility does not degrade immediately for PMC. Instead, we observe the strongest effects on utility later at the epoch where the model's distribution collapses for forget questions. We provide an ablation study on different lambda in Appendix A.3. We also find that gradient checkpointing can have negative effects on this utility drop and provide a more detailed ablation study in Appendix A.9.

**Faster approximation (PMC-fast).** We also implement a faster approximation making use of the result in Appendix A.6 that the reward can converge quickly. The main idea is to only select samples if their reward is larger than that of the sample with the best reward observed so far, and to stop sampling once the reward has converged to the maximum reward. This significantly reduces the number of samples during training, leading to faster runtimes in particular for Phi-1.5. We include this fast approximation as "PMC-fast". We further observe that Llama-3.2-3B-Instruct quickly collapses to generating EOS tokens (exclusively on the forget and paraphrased forget sets), significantly reducing runtime during sampling, and consequently the fast approximation does not lead to significant speed-ups for this model. For Gemma-3-12b-it, the fast approximation does not lead to significant speed-ups either since the reward does not converge to the maximum reward within the first 10 epochs in this setup, which is required for the fast approximation to be effective. Note that this is entirely model-, hyperparameter- and reward-dependent (e.g., the current reward does not penalize empty responses).

The results also indicate that the fast approximation can be harmful for utility, however, this discussion is more complicated as the results also depend on the hyperparameter selection (we use the same hyperparameters for both PMC and PMC-fast, and for a better comparison we would have to perform hyperparameter searches for both). Overall, please note that the broader unlearning problem is more complex and approximations affect not just runtime and utility/unlearn quality, but also other factors such as robustness. Deriving faster versions or approximations of PMC would require careful consideration of various design choices and hyperparameters, and may also introduce additional sources of error that need to be studied carefully. In this paper, we are more interested in studying the vanilla collapse process without additional sources of error, which is why we focus on the standard version of PMC in all other experiments, and consider efficiency as out of scope for this paper. In particular, note that one could even consider slowing down the collapse process intentionally, e.g. by choosing samples in a way that the reward increases more slowly, since a more thorough collapse process could have positive effects on robustness. We consider further improvements regarding the efficiency of collapse-based unlearning under such considerations as interesting future work.

**Limitations.** We do not aim to provide a complete estimate of computational costs and acknowledge that the measurements provided here can be influenced by various factors. In particular, note that all runtime experiments show individual runs running on a cluster simultaneously with other experiments, which may lead to variability in the reported runtimes (despite running all experiments on the same hardware). We consider the results as a first step to analyze the computational costs associated with fine-tuning LLMs on their own generations. A more detailed analysis of the computational costs of PMC-unlearning, including a breakdown of the costs associated with different components of the algorithm (e.g., sampling, optimization) would be interesting for future work.

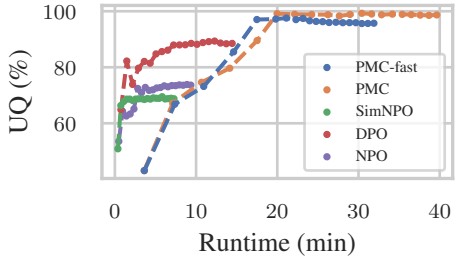 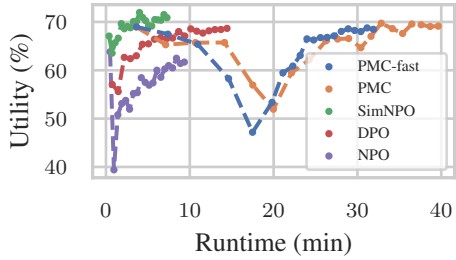

Figure 9: Runtime versus unlearning quality (left) and utility (right) for different unlearning methods (`Phi-1.5`). UQ: Unlearn quality. Points represent epochs (20 epochs in total).

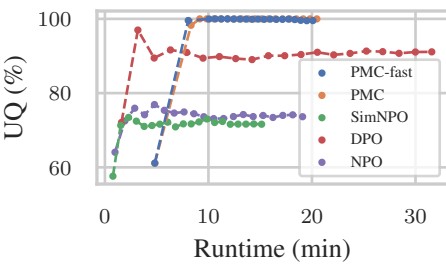 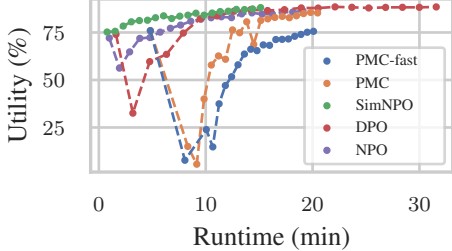

Figure 10: Runtime versus unlearning quality (left) and utility (right) for different unlearning methods (`Llama-3.2-3B-Instruct`). UQ: Unlearn quality. Points represent epochs (20 epochs in total).

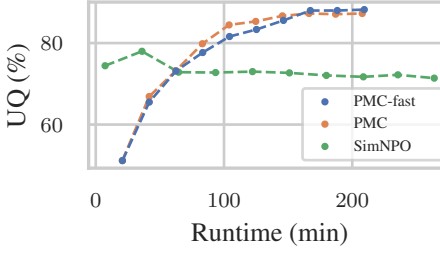 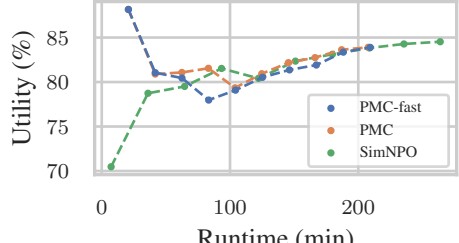

Figure 11: Runtime versus unlearning quality (left) and utility (right) for different unlearning methods (`Gemma-3-12b-it`). UQ: Unlearn quality. Points represent epochs (10 epochs in total).

## A.3 DETAILED ABLATIONS ON UTILITY-UNLEARNING TRADE-OFF PARAMETER

We provide a detailed ablation study on the utility-unlearning trade-off parameter $\lambda$ by analyzing unlearn quality and utility by epoch in Figure 12, Figure 13 and Figure 14, and by runtime in Figure 15, Figure 16 and Figure 17 for Phi-1.5, Llama-3.2-3B-Instruct and Gemma-3-12b-it, respectively. We generally observe that larger $\lambda$ leads to stronger utility at the cost of unlearn quality, as indicated by the formulation of PMC loss function.

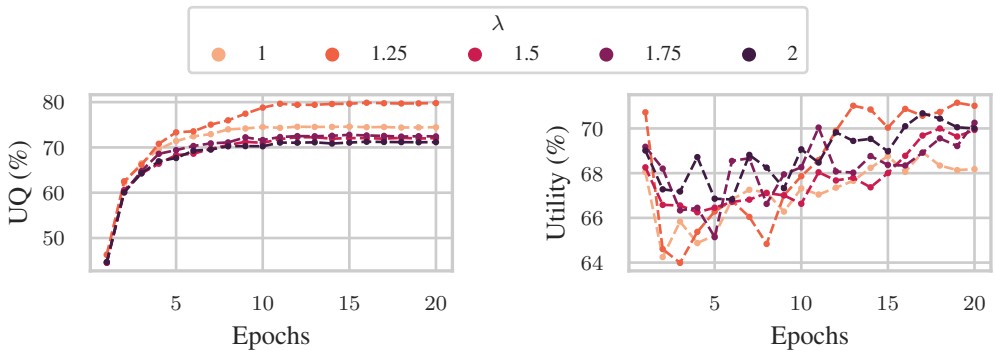

Figure 12: Unlearning quality (left) and utility (right) after each epoch of single runs with different $\lambda$ (`Phi-1.5`). UQ: Unlearn quality. Points represent epochs (20 epochs in total).

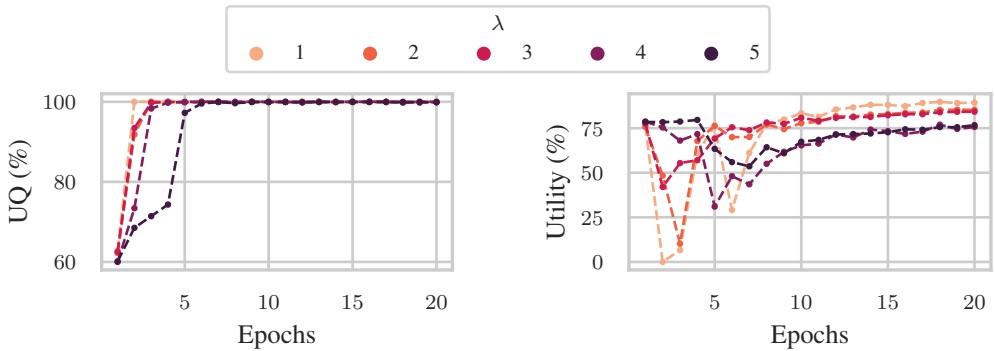

Figure 13: Unlearning quality (left) and utility (right) after each epoch of single runs with different $\lambda$ (`Llama-3.2-3B-Instruct`). UQ: Unlearn quality. Points represent epochs (20 epochs in total).

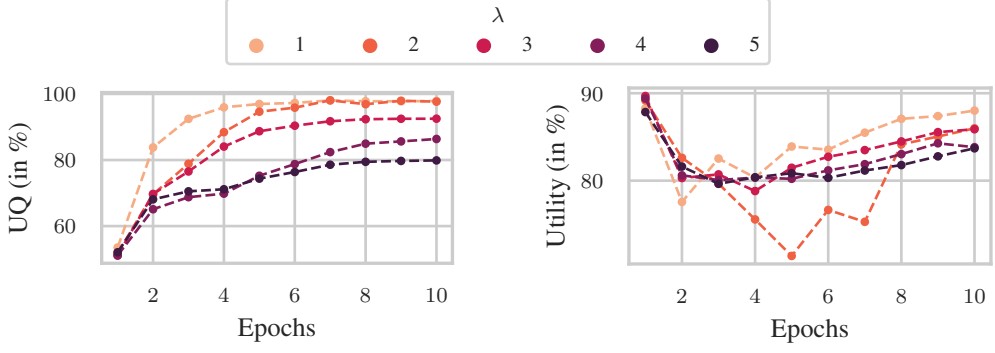

Figure 14: Unlearning quality (left) and utility (right) after each epoch of single runs with different $\lambda$ (`Gemma-3-12b-it`). UQ: Unlearn quality. Points represent epochs (10 epochs in total).

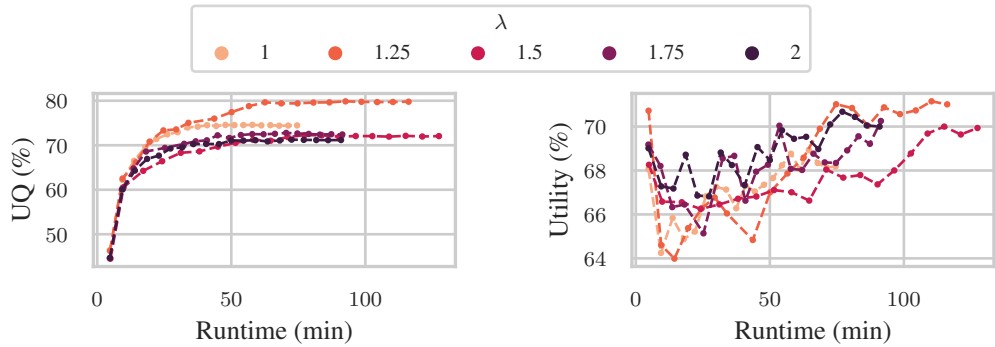

Figure 15: Single-run runtime versus unlearning quality (left) and utility (right) for different $\lambda$ (`Phi-1.5`). UQ: Unlearn quality. Points represent epochs (20 epochs in total).

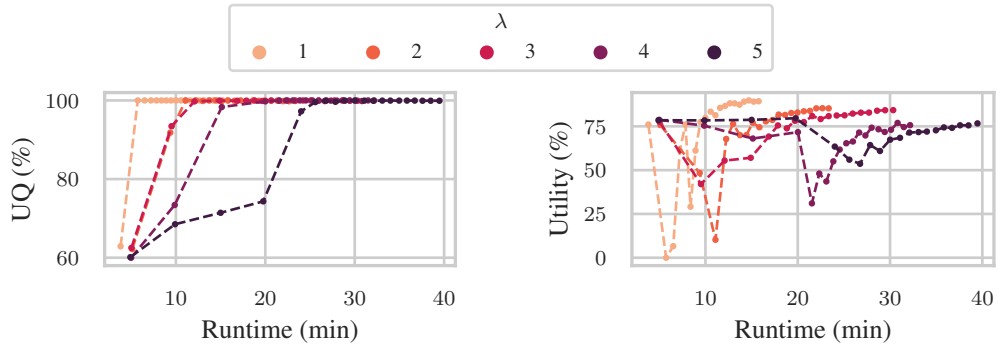

Figure 16: Single-run runtime versus unlearning quality (left) and utility (right) for different $\lambda$ (`Llama-3.2-3B-Instruct`). Points represent epochs (20 epochs in total).

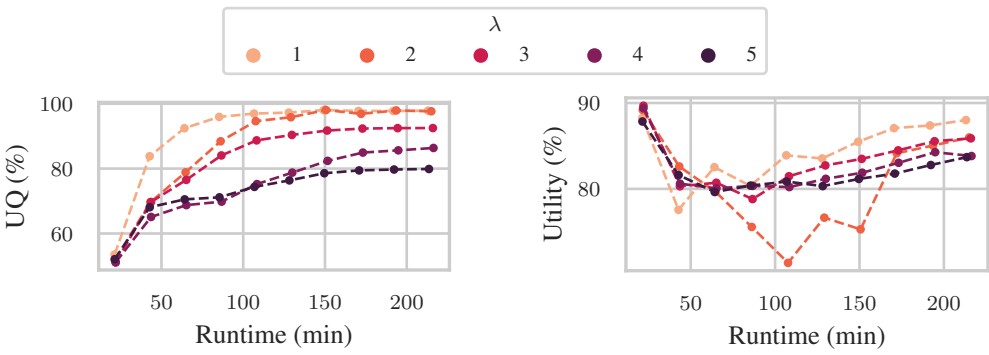

Figure 17: Single-run runtime versus unlearning quality (left) and utility (right) for different $\lambda$ (`Gemma-3-12b-it`). UQ: Unlearn quality. Points represent epochs (10 epochs in total).

## A.4 DETAILED ABLATIONS ON NUMBER OF SAMPLES

We provide a detailed ablation study on the number of samples $n$ by analyzing unlearn quality and utility by epoch in Figure 18, Figure 19 and Figure 20, and by runtime in Figure 21, Figure 22 and Figure 23 for Phi-1.5, Llama-3.2-3B-Instruct and Gemma-3-12b-it, respectively. We generally observe that more samples leads to stronger unlearn quality, however, the effects are more complex.

Phi-1.5 requires at least 10 samples to observe collapse to responses with maximum reward. Interestingly, even for 10 samples the collapse only happens in the 8th-epoch. For smaller number of samples, the process converges toward responses with suboptimal rewards (for which we would have to use a judge model to decide if the model unlearned the response). Clearly, more samples increase the probability of sampling responses with the maximum reward (see also discussion in Appendix A.6). Llama-3.2-3B-Instruct models typically collapse to generating EOS tokens for forget questions, which speeds up the sampling and unlearning process. The overall unlearning process takes longer if the collapse occurs at later epochs as e.g. in the case of a single sample (Figure 22).

Across all models we observe utility drops at the epoch where the distribution collapses for forget questions, from which the models quickly recover afterwards. We believe this indicates convergence towards a model that is not linearly mode connected to the original model, and we consider analyzing this phenomenon in more detail as an interesting direction for future work.

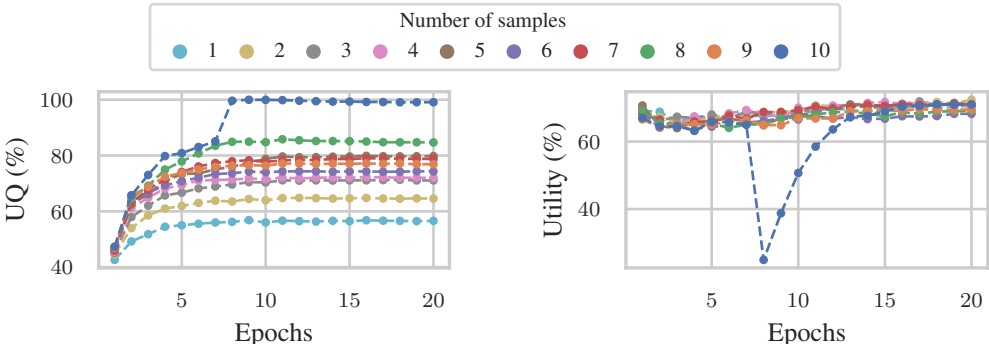

Figure 18: Unlearning quality (left) and utility (right) after each epoch of single runs with different number of samples $n$ (Phi-1.5). UQ: Unlearn quality. Points represent epochs (20 epochs in total).

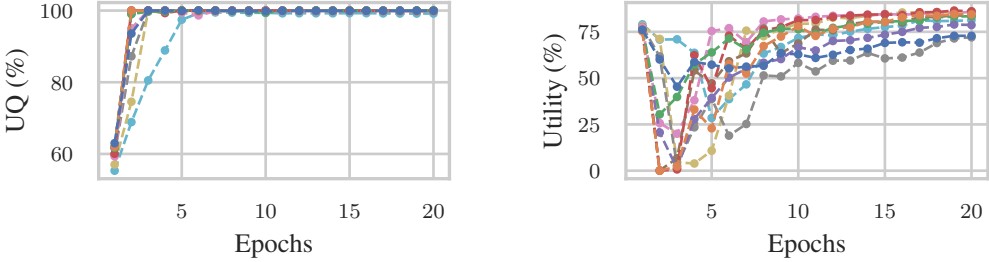

Figure 19: Unlearning quality (left) and utility (right) after each epoch of single runs with different number of samples $n$ (Llama-3.2-3B-Instruct). Points represent epochs (20 epochs in total).

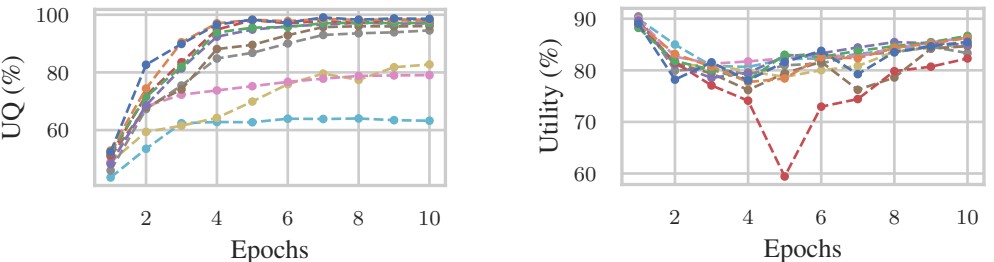

Figure 20: Unlearn quality (left) and utility (right) after each epoch of single runs with different number of samples $n$ (Gemma-3-12b-it). Points represent epochs (10 epochs in total).

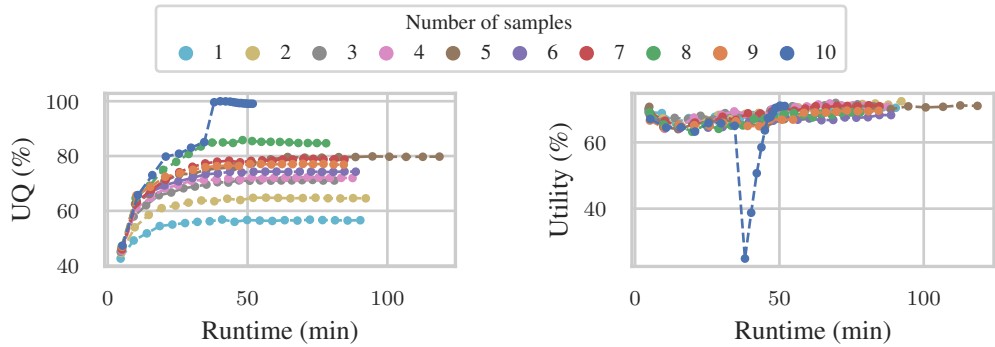

Figure 21: Single-run runtime versus unlearning quality (left) and utility (right) for different number of samples $n$ (`Phi-1.5`). UQ: Unlearn quality. Points represent epochs (20 epochs in total).

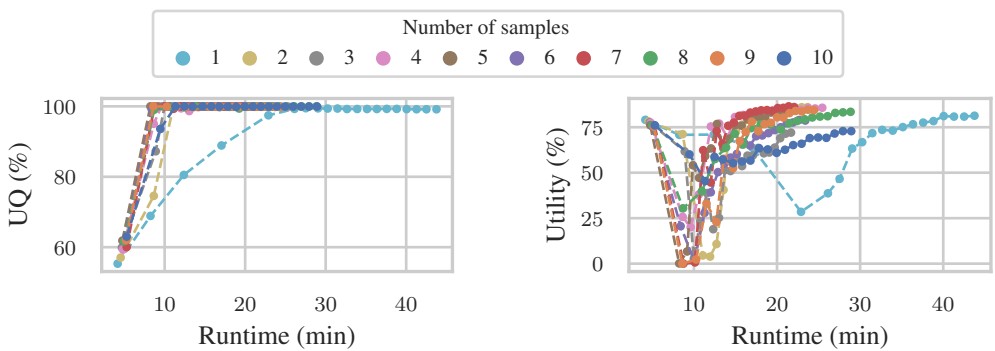

Figure 22: Single-run runtime versus unlearning quality (left) and utility (right) for different number of samples $n$ (`Llama-3.2-3B-Instruct`). Points represent epochs (20 epochs in total).

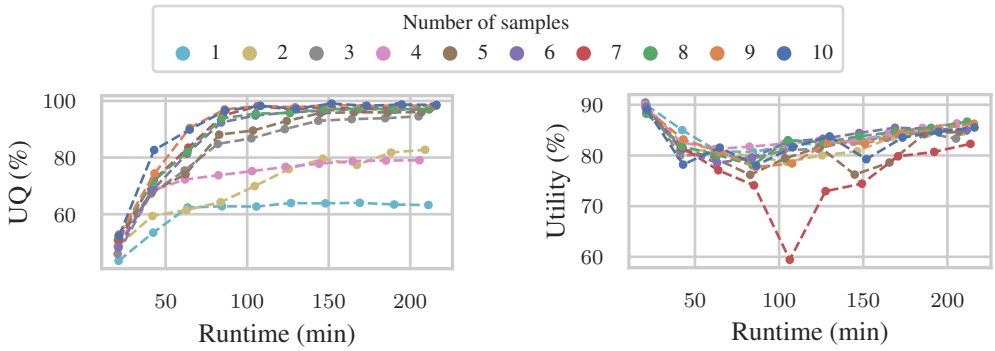

Figure 23: Single-run runtime versus unlearning quality (left) and utility (right) for different number of samples $n$ (`Gemma-3-12b-it`). Points represent epochs (10 epochs in total).

## A.5 ADDITIONAL SAMPLING AND PREFILLING ATTACK VISUALIZATION

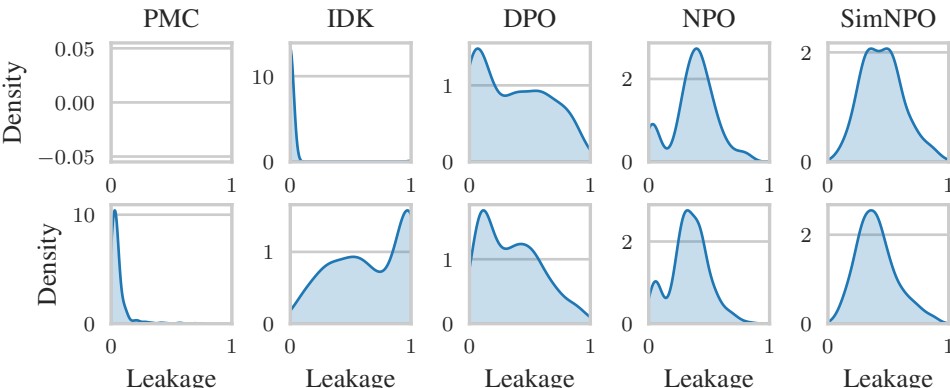

Figure 24: KDE-plot showing forget question distribution over worst-case leakage. Top-row shows the sampling attack, bottom-row prefilling+sampling attack. PMC is more robust against sampling and prefilling attacks. Lower worst-case leakage is better.

Figure 24 shows the distribution of forget questions according to their worst-case leakage (i.e. not just the average worst-case as in Figure 4). Interestingly, the IDK-baseline shows significant leakage for the majority of inputs only after the prefilling attack. DPO exhibits considerable leakage, with a few questions leaking the entire ground truth under sampling. PMC-unlearned models show significantly improved robustness against both sampling and prefilling attacks, with responses to forget questions having no similarity to the ground-truth answers for basically all forget questions.

## A.6 EMPIRICAL REWARD CONVERGENCE

We empirically analyze reward convergence from Theorem 1 in Figure 25. In particular, Figure 25 shows the batch-wise sample mean and variance of the reward during PMC-unlearning of Llama-3.2-3B-Instruct on the TOFU forget set over the first 500 training steps. We observe that the reward effectively converges to the maximum reward within the first 50 steps, and the variance vanishes.

For Llama-3.2-3B-Instruct we empirically observe this convergence to the maximum reward consistently across many different hyperparameter settings, as one can also see from the collapse evaluation in Figure 19. For Phi-1.5 and Gemma-3-12b-it, we also observe convergence to the maximum reward for certain hyperparameter settings, however, the convergence behavior can be more unstable across different hyperparameter settings, which may be due to maximum reward responses being more unlikely to be observed if the number of samples is too small for these models (which can lead to convergence to suboptimal rewards instead). Critically, for all models we found hyperparameter configurations for which the reward converges to the maximum reward empirically.

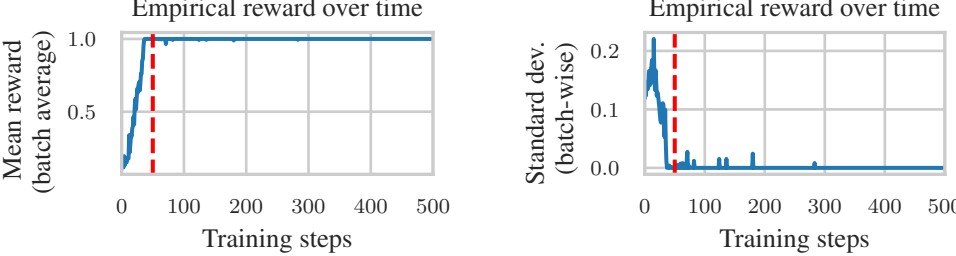

Figure 25: Empirically, the expected reward converges to the maximum reward in the first 50 training steps (25 steps are one epoch) and the variance vanishes. Vertical line shows step 50 (end of epoch 2).

**Model collapse with restart.** In separate experiments we also implement a version of PMC where we restart the collapse process after convergence to suboptimal rewards, i.e., we treat the unlearned model as a new model and repeat the process to diverge away from the newer model's outputs. This can cause another collapse later on (further improving unlearning quality), provided that the distribution is not fully collapsed yet. We leave such interesting cascaded collapses to future work.

## A.7 REWARD ABLATION STUDY

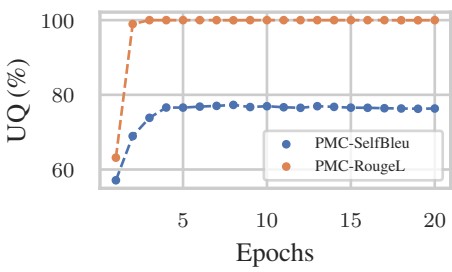 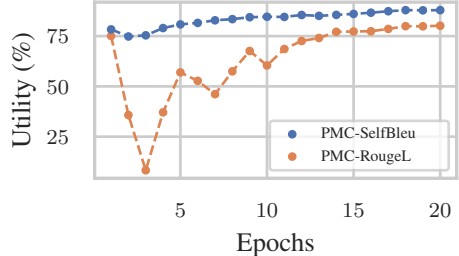

Figure 26: Reward ablation: Unlearning quality (left) and utility (right) after each epoch for different reward functions (`Llama-3.2-3B-Instruct`). UQ: Unlearn quality. Points represent epochs.

In the reward ablation study in Figure 26 we compare the standard ROUGE-L-based reward to a different reward based on self-BLEU scores, which measures the similarity between one generated samples to all other generated samples (and rewards samples that are most dissimilar to all others). In this setting we observe that using self-BLEU scores can lead to higher utility. Note that we would have to evaluate unlearning quality with a judge model to correctly evaluate if self-BLEU -based rewards are effective in unlearning, which cannot be concluded based on ROUGE-L scores in this case (although zero ROUGE-L implies successful unlearning in the sense that the output does not contain any words of ground-truth sequences, responses with higher scores may still not encode any ground-truth information). By manually investigating the resulting greedy generation after unlearning with self-BLEU rewards, we indeed observe successful semantic unlearning. This suggests that self-BLEU rewards can be effective for unlearning as well, although more thorough evaluations based on judge models would be important to further confirm this hypothesis.

## A.8 EXTENDED UTILITY EXPERIMENTS

We conducted additional experiments to test whether PMC introduces unexpected utility degradations on common benchmarks from the literature. Specifically, we compared the vanilla (base) models with their PMC-unlearned counterparts on Arc-Challenge, Arc-Easy (Clark et al., 2018), and MMLU (Hendrycks et al., 2021). We report the mean and standard deviation over five random seeds for Phi and Llama. The results in Table 1 show that PMC has only minor impact on model utility. Note that models can already degrade in utility during the initial fine-tuning on TOFU-full, and that the utility remains high, independent of whether we apply the chat-template used during TOFU fine-tuning (results in Table 1 are without applying the TOFU chat-template during evaluation).

Table 1: Model utility comparison between base models and models fine-tuned with PMC.

|  | **Arc-Challenge** | **Arc-Easy** | **MMLU** |
|---|---|---|---|
| `Phi-1.5` (base) | 0.4462 | 0.7622 | 0.4174 |
| `Phi-1.5` (PMC) | $0.4283 \pm 0.0057$ | $0.6891 \pm 0.0053$ | $0.4063 \pm 0.0031$ |
| `Llama-3.2-3B-Instruct` (base) | 0.4368 | 0.7382 | 0.6041 |
| `Llama-3.2-3B-Instruct` (PMC) | $0.4341 \pm 0.0061$ | $0.7230 \pm 0.0058$ | $0.5924 \pm 0.0040$ |
| `Gemma-3-12b-it` (base) | 0.6083 | 0.8354 | 0.7151 |
| `Gemma-3-12b-it` (PMC) | 0.6032 | 0.8421 | 0.6934 |

### A.9 GRADIENT CHECKPOINTING ABLATION

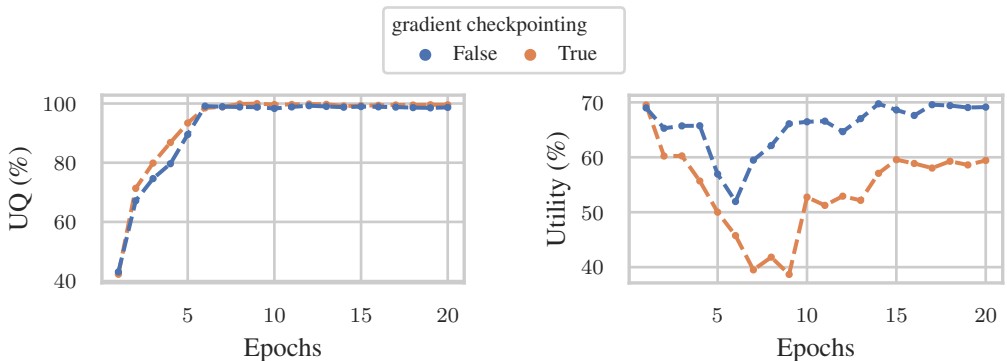

Figure 27: Gradient checkpointing ablation: Unlearning quality (left) and utility (right) after each epoch for `Phi-1.5`. UQ: Unlearn quality. Points represent epochs.

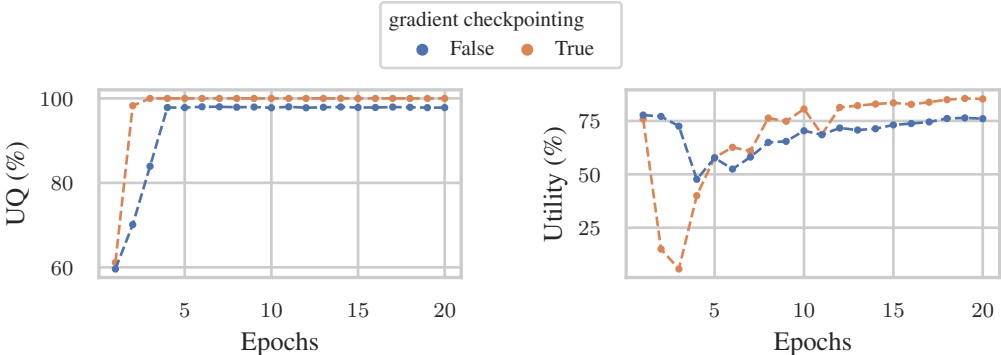

Figure 28: Gradient checkpointing ablation: Unlearning quality (left) and utility (right) after each epoch for `Llama-3.2-3B-Instruct`. UQ: Unlearn quality. Points represent epochs.

We use gradient checkpointing in all our Llama-3.2-3B-Instruct and Gemma-3-12b-it experiments as this corresponds to the setup of the initial TOFU repository (Maini et al., 2024). While gradient checkpointing should in principle just use compute to save memory (to fit larger models), we found it can introduce an additional source of error. In particular, we observe entirely different unlearning trajectories for fixed hyperparameters and seeds (see ablations in Figure 27 and Figure 28).

We advocate for more careful experimental setups in future unlearning research, as such unnecessary sources of error can prevent robust conclusions. We recommend to either avoid these techniques or to carefully ablate them to ensure their effects are well understood and results are reliable. Critically, since we perform experiments without (Phi) and with gradient checkpointing (Llama and Gemma), we believe the main conclusions of our work are robust since we observe the same general trends across all models and many hyperparameter settings (even if the exact unlearning trajectories may vary).

A.10 ADDITIONAL EVALUATION ON MUSE-NEWS DATA

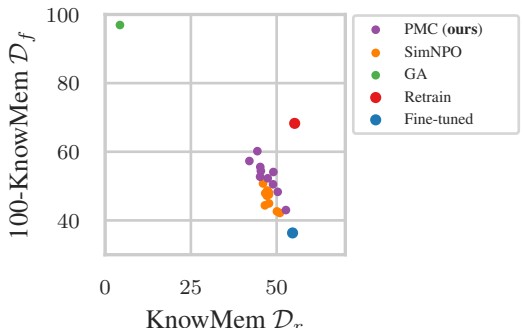

Figure 29: Unlearning-utility trade-offs on MUSE-news subset.

**Experimental setup.** For this experiment we consider a subset of the MUSE-news dataset (Shi et al., 2025) (which we call MUSE-news-small) as follows: We construct the retain and forget texts by keeping only those sentences whose tokens overlap with answers in forget and retain questions, respectively. This results in a reduced MUSE-news dataset consisting of 454 forget and 1,714 retain sentences. Note that this subset still represents a reasonable dataset for evaluating machine unlearning approaches, in particular because existing state-of-the-art unlearning methods still fail to unlearn even on this reduced dataset. We train all methods for 10 epochs with learning rate $1e^{-5}$. We set $\lambda$ to 1.0 for all experiments. We chose SimNPO as baseline because it dominates all other prior approaches on this dataset (Fan et al., 2024). For the SimNPO baseline we set $\beta$ to 0.7 and $\gamma$ to 0.0 as suggested in Fan et al. (2024). We compute all unlearning methods on sentence-level. All other hyperparameters remain the same as described by the dataset (Shi et al., 2025). As model we use the Llama-2-7b-hf model fine-tuned on MUSE-news as provided by Shi et al. (2025). We evaluate 10 checkpoints for each method (except for GA, which already shows poor performance after the first epoch). For evaluation we follow the exact evaluation protocol described by Shi et al. (2025) to compute KnowMem scores for forget and retain evaluations.

**Collapse-based unlearning beyond Q&A.** We implement our approach for the MUSE-news dataset as follows: We draw random positions in each sentence and prefill the preceding tokens. We then generate continuations to fill the rest of the sentence by sampling from the model at the current iteration using temperature of 1 and top-p of 0.95. In our experiments we use 8 random positions and draw 5 candidates per position. We then fine-tune on the continuations selected using the same reward function as for our TOFU experiments.

**Outcome.** Figure 29 shows that PMC can expand the Pareto-front w.r.t. utility and unlearning KnowMem metrics when evaluated on MUSE-news-small.

**Discussion.** Despite the initial positive results, we do not believe that it is possible to draw robust conclusions using the MUSE benchmark: While unlearning is performed on a rather large collection of sentences, the evaluation itself covers only a very small subset of those sentences (and even for those sentences we found the evaluation rather insufficient). We believe that in machine unlearning, experimental setups should thoroughly evaluate if models unlearned the information that they should forget during unlearning. While the task studied in MUSE is interesting conceptually (since it goes beyond Q&A tasks), we found the overall evaluation protocol insufficient for our analysis. We advocate for more careful and targeted experimental setups in future unlearning research.

**Take-away.** We consider our MUSE-experiments only as a pilot study to demonstrate how to extend collapse-based unlearning to tasks beyond Q&A and do not draw any conclusions based on these outcomes. We consider the development of improved experimental setups and evaluations for unlearning beyond Q&A as an interesting direction for future work.

## B  FULL EXPERIMENTAL SETUP AND IMPLEMENTATION DETAILS

We conduct all experiments on NVIDIA A100 GPUs (40GB), NVIDIA H100 GPUs (80GB) and NVIDIA H200 GPUs (140 GB). We specify hardware details regarding specific plots in Appendix B.4. We provide source code for all experiments via the project page.

**Datasets.** We use the TOFU Q&A dataset (Maini et al., 2024) for finetuning and unlearning. The dataset consists of 4,000 question-answer pairs about generated autobiographies of 200 different, fictitious authors. We use the "forget10" split of the dataset, since it is the most challenging split of the dataset. The split uses 400 samples for the forget set and the remaining samples for the retain set. To facilitate model evaluation we approximate retain performance using a (random but fixed) subset of 400 retain samples.

### B.1  FINE-TUNING DETAILS

We fine-tune three pretrained LLMs, Phi-1.5 (Li et al., 2023) and Llama-3.2-3B-Instruct (Grattafiori et al., 2024), and Gemma-3-12b-it (DeepMind, 2025). We generally follow the experimental setup described in (Maini et al., 2024), and fine-tune models on the full TOFU Q&A dataset.

**Fine-tuning hyperparameters.** We fine-tune both models for 5 epochs using the AdamW optimizer (Loshchilov & Hutter, 2019) together with ZeRO-3 (Rajbhandari et al., 2020). For fine-tuning Phi-1.5 we use a batch size of 16 and gradient accumulation steps of 2, which results in an effective batch size of 32. For Llama-3.2-3B-Instruct and Gemma-3-12b-it we use a batch size of 8 and gradient accumulation steps of 2, which results in an effective batch size of 16. We use a learning rate of $2e^{-5}$ for Phi-1.5 and $1e^{-5}$ for Llama-3.2-3B-Instruct and Gemma-3-12b-it. We also apply weight decay of 0.01 for all models. For Llama-3.2-3B-Instruct and Gemma-3-12b-it we additionally deploy gradient checkpointing (Chen et al., 2016) and disable flash attention 2. We summarize hyperparameters in Table 2 and results in Table 3.

Table 2: Fine-tuning hyperparameters for Phi-1.5, Llama-3.2-3B-Instruct and Gemma-3-12b-it.

| Fine-tuning hyperparameter | Phi-1.5 | Llama-3.2-3B-Instruct | Gemma-3-12b-it |
|---|---|---|---|
| Batch size | 16 | 8 | 8 |
| Gradient accumulation steps | 2 | 2 | 2 |
| Learning rate | 2e-5 | 1e-5 | 1e-5 |
| Number of epochs | 5 | 5 | 5 |
| Weight decay | 0.01 | 0.01 | 0.01 |
| Gradient checkpointing | False | True | True |

Table 3: ROUGE-L scores as well as unlearn quality (UQ) and utility (as defined in Section 5) for the pretrained models (before fine-tuning) and the models after fine-tuning on the TOFU 90/10 split.

| Model | Full | World-facts | Real-authors | Forget | Paraph. forget | Retain | UQ | Utility |
|---|---|---|---|---|---|---|---|---|
| `Phi-1.5` | 0.45 | 0.82 | 0.66 | 0.45 | 0.39 | 0.45 | 0.58 | 0.64 |
| `Phi-1.5` (FT) | $0.93 \pm 0.00$ | $0.75 \pm 0.02$ | $0.44 \pm 0.01$ | $0.92 \pm 0.00$ | $0.31 \pm 0.00$ | $0.91 \pm 0.01$ | $0.38 \pm 0.00$ | $0.70 \pm 0.01$ |
| `Llama-3.2-3B-I.` | 0.26 | 0.92 | 0.96 | 0.27 | 0.24 | 0.26 | 0.75 | 0.71 |
| `Llama-3.2-3B-I.` (FT) | $0.96 \pm 0.00$ | $0.90 \pm 0.01$ | $0.86 \pm 0.02$ | $0.95 \pm 0.00$ | $0.34 \pm 0.00$ | $0.96 \pm 0.00$ | $0.35 \pm 0.00$ | $0.91 \pm 0.01$ |
| `Gemma-3-12b-it` | 0.38 | 0.95 | 0.99 | 0.38 | 0.30 | 0.37 | 0.66 | 0.77 |
| `Gemma-3-12b-it` (FT) | $0.99 \pm 0.00$ | $0.92 \pm 0.02$ | $0.93 \pm 0.01$ | $0.99 \pm 0.00$ | $0.40 \pm 0.00$ | $0.99 \pm 0.00$ | $0.31 \pm 0.00$ | $0.95 \pm 0.01$ |

## B.2 UNLEARNING DETAILS

For the unlearning experiments we use the same hyperparameters as for fine-tuning, except when stated otherwise in the following grid search. For a fair comparison between methods, we run 100 experiments for each method. We repeat each experiment 5 times using the same fixed random seeds for all methods and report mean across the runs. That is we run 500 experiments for each method. We summarize the hyperparameters used for the grid search in Table 4. Note that we introduce $\lambda$ as a trade-off between retain and forget loss for all methods, even if their original formulation does not include it. For all experiments we use the vanilla implementation of PMC (not PMC-fast). We provide all configurations along with the source code to facilitate reproducibility and future research.

Table 4: Gridsearch details for all unlearning methods. For a fair comparison, we run 500 experiments for each method: 100 different configurations each repeated for 5 different seeds. LR: Learning rate.

| Parameter | GA | | Parameter | GD | | Parameter | IDK |
|---|---|---|---|---|---|---|---|
| Seed | range(0,5) | | Seed | range(0,5) | | Seed | range(0,5) |
| LR | linspace(1e-5, 1e-4, 10) | | LR | {1e-5, 2e-5} | | LR | {1e-5, 2e-5} |
| Epochs | linspace(2, 20, 10) | | Epochs | {3, 5, 10, 15, 20} | | Epochs | {3, 5, 10, 15, 20} |
| | | | $\lambda$ | linspace(0.5, 1.5, 10) | | $\lambda$ | linspace(0.5, 1.5, 10) |

| Parameter | DPO | | Parameter | NPO | | Parameter | SimNPO |
|---|---|---|---|---|---|---|---|
| Seed | range(0,5) | | Seed | range(0,5) | | Seed | range(0,5) |
| LR | 1e-05 | | LR | 1e-05 | | LR | 1e-05 |
| Epochs | 10 | | Epochs | 10 | | Epochs | 10 |
| $\lambda$ | linspace(0.5, 1.5, 10) | | $\lambda$ | linspace(0.5, 1.5, 10) | | $\lambda$ | linspace(0.05, 0.25, 4) |
| $\beta$ | linspace(0.05, 0.2, 10) (includes $\beta = 0.1$) | | $\beta$ | linspace(0.05, 0.2, 10) (includes $\beta = 0.1$) | | $\beta$ | linspace(2.5, 5.5, 5) |
| | | | | | | $\gamma$ | linspace(0.0, 2.0, 5) |

| Parameter | PMC (Phi-1.5) | | Parameter | PMC (Llama-3.2) | | Parameter | PMC (Gemma-3) |
|---|---|---|---|---|---|---|---|
| Seed | range(0,5) | | Seed | range(0,5) | | Seed | range(0,5) |
| LR | 1e-05 | | LR | 1e-05 | | LR | 1e-05 |
| Epochs | {10, 15} | | Epochs | {15, 20} | | Epochs | {15, 20} |
| $\lambda$ | linspace(0.5, 1.5, 5) | | $\lambda$ | {0.5, 0.75, 1, 2, 3} | | $\lambda$ | {0.5, 0.75, 1, 2, 3} |
| #Samples | {1, 5, 10, 15, 20} | | #Samples | {10, 15} | | #Samples | {10, 15} |
| Temperature | {1.25, 1.5} | | Temperature | {0.8, 0.9, 1, 1.25, 1.5} | | Temperature | {0.8, 0.9, 1, 1.25, 1.5} |
| Top-p | 0.95 | | Top-p | 0.95 | | Top-p | 0.95 |

### B.3 DETAILS ON PHI ABLATION STUDIES

Table 5: Overview of hyperparameters used for the ablation studies in Section 5 and Appendix A. For each setting, we repeat each experiments for 5 different seeds and report mean and standard deviation. All ablation studies in the main text are conducted with the Phi-1.5 model.

Number of epochs

| Parameter | DPO |
| --- | --- |
| Seed | range(0,5) |
| LR | 1e-05 |
| Epochs | range(1,21) |
| $\lambda$ | 1.5 |
| $\beta$ | 0.1 |

Number of epochs

| Parameter | NPO |
| --- | --- |
| Seed | range(0,5) |
| LR | 1e-05 |
| Epochs | range(1,21) |
| $\lambda$ | 1.5 |
| $\beta$ | 0.05 |

Number of epochs

| Parameter | SimNPO |
| --- | --- |
| Seed | range(0,5) |
| LR | 1e-05 |
| Epochs | range(1, 21) |
| $\lambda$ | 0.25 |
| $\beta$ | 4 |
| $\gamma$ | 0 |

Number of epochs

| Parameter | PMC |
| --- | --- |
| Seed | range(0,5) |
| LR | 1e-05 |
| Epochs | range(1, 21) |
| $\lambda$ | 1.5 |
| #Samples | 5 |
| Temperature | 1.25 |
| Top-p | 0.95 |

Number of samples

| Parameter | PMC |
| --- | --- |
| Seed | range(0,5) |
| LR | 1e-05 |
| Epochs | 15 |
| $\lambda$ | 1.5 |
| #Samples | range(1,21) |
| Temperature | 1.25 |
| Top-p | 0.95 |

$\lambda$

| Parameter | PMC |
| --- | --- |
| Seed | range(0,5) |
| LR | 1e-05 |
| Epochs | 15 |
| $\lambda$ | range(0.5, 1.55, 0.1) |
| #Samples | 5 |
| Temperature | 1.25 |
| Top-p | 0.95 |

Sampling temperature

| Parameter | PMC |
| --- | --- |
| Seed | range(0,5) |
| LR | 1e-05 |
| Epochs | 15 |
| $\lambda$ | 1.5 |
| #Samples | 5 |
| Temperature | range(1.0, 1.55, 0.1) |
| Top-p | 0.95 |

Top-p sampling

| Parameter | PMC |
| --- | --- |
| Seed | range(0,5) |
| LR | 1e-05 |
| Epochs | 15 |
| $\lambda$ | 1.5 |
| #Samples | 5 |
| Temperature | 1.25 |
| Top-p | {0.9, 0.95, 1} |

BT temperature $\tau$

| Parameter | PMC |
| --- | --- |
| Seed | range(0,5) |
| LR | 1e-05 |
| Epochs | 15 |
| $\lambda$ | 1.5 |
| #Samples | 5 |
| Temperature | 1.25 |
| Top-p | 0.95 |
| $\tau$ | linspace(0.05, 1, 20) |

### B.4  HYPERPARAMETER DETAILS FOR ADDITIONAL EXPERIMENTS

All experiment configurations are released together with the code. We use the vanilla implementation of PMC (not PMC-fast) for all experiments, unless explicitly stated otherwise. Note that experiments described in this section are not averaged over multiple seeds, they are results of single-seed runs. Unless stated otherwise, all experiments in the main paper (including Figure 5) are performed with Phi-1.5 except for the sampling analysis, for which we use Llama-3.2-3B-Instruct. Ablation studies for Llama-3.2-3B-Instruct and Gemma-3-12b-it are reported in Appendix A.3 and Appendix A.4.

**Hardware details.** The runtime experiments in Appendix A.2 are performed on a single H200 GPU. Runtime experiments in Appendix A.3 and Appendix A.4 are performed on a single A100 GPU for Phi-1.5 and a single H200 GPU for Llama-3.2-3B-Instruct and Gemma-3-12b-it. All other experiments reported in the paper use A100s, H100s or H200s based on cluster availability.

**Hyperparameters for computational cost analysis (Appendix A.2).** Unless stated differently, we use the exact experimental setup described in Appendix B. We start describing the hyperparameters used for Phi-1.5: For vanilla PMC and PMC-fast we use $\lambda = 1$, top-p of 0.95, temperature of 1.25, and 5 samples. For SimNPO we use $\lambda = 0.25$, $\beta = 4$, and $\gamma = 0$, and for NPO we use $\lambda = 1.5$ and $\beta = 0.05$. For Llama-3.2-3B-Instruct and Gemma-3-12b-it we choose $\lambda = 1.25$ for PMC and PMC-fast. All other hyperparameters remain as described before. We train Phi-1.5 and Llama-3.2-3B-Instruct for 20 epochs and Gemma-3-12b-it for 10 epochs.

**Hyperparameters for $\lambda$-ablation (Appendix A.3).** For all models we chose $\lambda$ as described in the plots and use top-p of 0.95, temperature of 1, and 10 samples. We train Phi-1.5 and Llama-3.2-3B-Instruct for 20 epochs and Gemma-3-12b-it for 10 epochs.

**Hyperparameters for samples-ablation (Appendix A.4).** We set $\lambda = 1.25$ for Phi-1.5, and $\lambda = 1.5$ for Llama-3.2-3B-Instruct and Gemma-3-12b-it. For all models we use top-p of 0.95, and temperature of 1. The number of samples is varied as described in the plots. We train Phi-1.5 and Llama-3.2-3B-Instruct for 20 epochs and Gemma-3-12b-it for 10 epochs.

**Hyperparameters for sampling and prefilling attack (Appendix A.5).** For the sampling and prefilling experiment (Figure 4 and Appendix 5.2) we train Llama-3.2-3B-Instruct models with five different unlearning techniques (PMC, IDK, NPO, SimNPO, DPO) with the following hyperparameters: For all methods we use a learning rate of $1e^{-5}$ and 20 epochs. For PMC and IDK we choose $\lambda = 1.25$. For PMC we use 20 samples, a temperature of 0.9, and top-p of 0.95 during PMC sampling. For NPO we use $\lambda = 1.5$ and $\beta = 0.05$ and for DPO $\beta = 0.1$. For SimNPO we use $\lambda = 0.25$, $\beta = 4$, and $\gamma = 0$. All other hyperparameters remain as described in Appendix B.

After unlearning, we sample 100 responses per question from each model using a temperature of 0.9 and top-p of 0.95. We then compute the ROUGE-L score between each sampled response and the ground-truth answer. We report the worst-case ROUGE-L score average over all questions in the forget set and report it in Figure 4 in the main paper. The worst-case ROUGE-L score for a question is defined as the maximal ROUGE-L score across all 100 sampled responses for that question. Figure 24 shows a KDE-plot of ROUGE-L scores over all samples.

**Hyperparameters for reward convergence ablation (Appendix A.6)**. We run PMC-unlearning for 20 epochs with learning rate $1e^{-5}$, $\lambda = 1.25$, 20 samples, temperature of 0.9, and top-p of 0.95. Model used for the plot in Figure 25 is Llama-3.2-3B-Instruct.

**Hyperparameters for reward ablation (Appendix A.7).** We perform the experiment with Llama-3.2-3B-Instruct using $\lambda = 1$, top_p=0.95, temperature=1.25, num_samples=10, num_epochs=20, and use the PMC-fast implementation.

**Hyperparameters for gradient checkpointing ablation (Appendix A.9).** For Phi-1.5 we use $\lambda = 1$, top-p of 0.95, temperature of 1.25, and 5 samples. For Llama-3.2-3B-Instruct we use $\lambda = 1.25$, top-p of 0.95, temperature of 0.9, and 5 samples. We train both models for 20 epochs.

### B.5 MPC PROMPT TEMPLATE

We created the MPC dataset from the TOFU Q&A dataset by prompting ChatGPT to do this specific task. We selected a subset of 84 questions based on their suitability to be converted to a multiple choice format. Suitability was evaluated using ChatGPT with the following template:

---

Answer with either "Yes" if the following is a factual question e.g., it can be answered with a few words, such as names, dates, orientation, etc., or "No" if it requires longer explanations. Do not output anything beyond "Yes", or "No".

**QUESTION:** {question}
**ANSWER:** {answer}

---

The prompt template used to convert the dataset to MPC is shown in the following:

---

Convert the following question and answer into a multiple choice question with 4 possible answers. For each option remain close to the original sentence structure. Here is an example of an original question and answer:

**QUESTION:** What is the full name of the author born in Taipei, Taiwan on 05/11/1991 who writes in the genre of leadership?
**ANSWER:** The author's full name is Hsiao Yun-Hwa.

What should be generated in this case:

**MPC ANSWER:**
A) The author's full name is Hsiao Yun-Hwa.
B) The author's full name is Ming-Chi Lee
C) The author's full name is Wei-Li Chen
D) The author's full name is Yu-Ting Huang

**CORRECT ANSWER:** A Do it for the following pair:

**QUESTION:** {question}
**ANSWER:** {answer}

**MPC ANSWER:**

---

## C  WARM-UP: ITERATIVE UNLEARNING WITH CATEGORICAL DISTRIBUTIONS

**Definition C.1** (Categorical distribution). A categorical distribution is a probability distribution over $K$ different possible outcomes $\{0, \ldots, K-1\}$ and parametrized by a vector $\pi = (p_0, \ldots, p_{K-1})$ of probabilities for each category, where $p_k \geq 0$ and $\sum_{k=0}^{K-1} p_k = 1$. The probability mass function is given by $\Pr[X = k] = p_k$.

**Definition C.2** (Model collapse). A random variable X is said to have a *collapsed* distribution if its variance is zero, i.e. $\mathrm{Var}[X] = 0$.

**Learning a categorical distribution.** Consider a random variable $X$ equipped with a categorical distribution over $K$ categories. We can learn the parameters $\pi$ of the distribution of $X$ from $n$ realizations $x = (x_1, \ldots, x_n)$ using maximum likelihood estimation (MLE). The likelihood function is given by

$$L(x; \pi) \triangleq \prod_{i=1}^{n} \Pr[X = x_i] = \prod_{k=0}^{K-1} p_k^{n_k} = \left(1 - \sum_{k=0}^{K-2} p_k\right)^{n_{K-1}} \prod_{k=0}^{K-2} p_k^{n_k}$$

where $n_k = \sum_{i=1}^{n} \mathbb{1}[x_i = k]$ is the number of times category $k$ was observed, and in the last equation we rewrote $p_{K-1}$ by making use of the fact that $\sum_{k=0}^{K-1} p_k = 1$. We maximize the log-likelihood function as follows:

$$\frac{\partial \log L(x; \pi)}{\partial p_k} = \frac{n_k}{p_k} - \frac{n_{K-1}}{1 - \sum_{i=0}^{K-2} p_i} \overset{!}{=} 0$$

$$\Leftrightarrow \quad p_k = \frac{n_k}{n_{K-1}} \left(1 - \sum_{i=0}^{K-2} p_i\right) \quad \Leftrightarrow \quad p_k + \frac{n_k}{n_{K-1}} \sum_{i=0}^{K-2} p_i = \frac{n_k}{n_{K-1}},$$

which is a linear system of $K-1$ equations. We can briefly verify that the solution to this linear system is given by $\hat{p}_k = \frac{n_k}{n}$:

$$\frac{n_k}{n} + \frac{n_k}{n_{K-1}} \sum_{i=0}^{K-2} \frac{n_i}{n} = \frac{n_k}{n_{K-1}} \left(\frac{n_{K-1}}{n} + \frac{\sum_{i=0}^{K-2} n_i}{n}\right) = \frac{n_k}{n_{K-1}}.$$

That is the MLE for the probability $p_k$ of category $k$ is the fraction $\frac{n_k}{n}$ of observing category $k$ among all $n$ samples.

**Iterative relearning categorical distributions.** Given an arbitrary categorical distribution with parameters $\pi_0$, we analyze iterative relearning of a categorical distribution on its own generated data. First we draw $n$ samples $x = (x_1, \ldots, x_n)$ i.i.d. from the distribution given by $\pi_0$. We then relearn the parameters $\pi_1$ from the dataset $x$ via maximum likelihood estimation. Repeating this process will lead to convergence as we show in the following:

**Proposition C.3.** *Iteratively relearning of a categorical distribution $\pi_t$ on its own generated data yields model collapse independent of the initial distribution.*

Intuitively, given finite samples, the iterative relearning process describes an absorbing Markov chain, which is known to converge to an absorbing state (Shumailov et al., 2023).

*Full proof.* For the sake of exposition we first consider the case of a categorical distribution with $K = 2$ categories, i.e. a Bernoulli distribution with a single success parameter $p$. Without loss of generality we further assume that the initial success probability is already a multiple of $\frac{1}{n}$ (otherwise just relearn once and then follow the proof).

The main idea of this proof is to model the stochastic process of relearning on self-generated data as a discrete-time discrete-state-space Markov chain. Specifically, during iterative relearning, the maximum likelihood estimate (average number of successes) itself becomes a random variable that defines the parameter for the distribution of the next iteration. We denote the number of successes in the $(t+1)$-th iteration as $Y_{t+1} = \sum_{i=1}^{n} X_t^{(i)}$, where $X_t^{(i)} \sim \mathrm{Ber}\left(\frac{Y_t}{n}\right)$ are i.i.d. Bernoulli random variables with success probability $\frac{Y_t}{n}$.

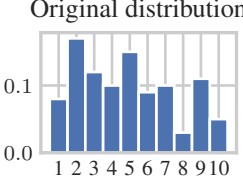 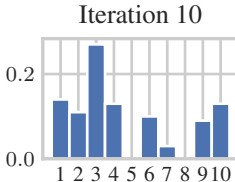 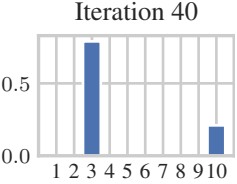 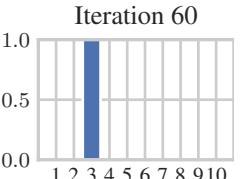

Figure 30: Model collapse for iterative retraining with categorical distributions. After 60 iterations, the distribution collapsed to a zero-variance distribution, i.e. a single category. Compare to Figure 2.

Note that there are only $n + 1$ possible Bernoulli distributions because we estimate the success probability with a discrete value. Thus the stochastic process of iterative relearning can be described as a Markov chain with state space $\{0, 1, \ldots, n\}$ corresponding to the $n + 1$ possible Bernoulli distributions. We further describe the stochastic process using a $(n + 1) \times (n + 1)$ transition matrix $P_n = (p_{ij})$ of the probabilities to transition from one distribution to another:

$$p_{ij} \triangleq \Pr\left[Y_{t+1} = j \mid Y_t = i\right] = \text{Binom}\left(j;\ n, i/n\right).$$

In other words, the rows of the transition matrix corresponds to the PMF of the Binomial distribution with $n$ samples and success probability $i/n$, where $i$ corresponds to the number of successes of the previous iteration.

As an example, we show transition matrices for $n = 1, 2, 3$ samples:

$$P_1 = \begin{bmatrix} 1 & 0 \\ 0 & 1 \end{bmatrix} \qquad P_2 = \begin{bmatrix} 1 & 0 & 0 \\ 1/4 & 1/2 & 1/4 \\ 0 & 0 & 1 \end{bmatrix}$$

$$P_3 = \begin{bmatrix} 1 & 0 & 0 & 0 \\ 0.29629630 & 0.44444444 & 0.22222222 & 0.03703704 \\ 0.03703704 & 0.22222222 & 0.44444444 & 0.29629630 \\ 0 & 0 & 0 & 1 \end{bmatrix}$$

Notably, the described Markov chain is a so-called absorbing Markov chain: First, it contains absorbing states ($0$ and $n$) corresponding to Bernoulli distributions with success probability zero or one – once a random walker reaches one of the absorbing states, the walker cannot leave it anymore. Second, it is possible to go from any transient (non-absorbing) state to an absorbing state in a finite number of steps. Thus a random walker is guaranteed to eventually reach an absorbing state, independent of the initial success probability.

Consequently, iterative relearning will result w.p.1 in a distribution with success probability zero or one. Since the variance of a Bernoulli distribution is $p(1 - p)$, the variance of the final distribution is zero, i.e., the distribution collapsed.

For the general case of a categorical distribution with $K$ categories, the proof follows analogously by considering the Markov chain with states corresponding to the possible $\binom{n+K-1}{K-1}$ categorical distributions $\mathbf{p}$. In this case, the rows correspond to the PMF of a Multinomial distribution: $p_{ij} = \text{Multinom}\left(n\mathbf{p}[j];\ n, \mathbf{p}[i]\right)$, where $\mathbf{p}[i]$ denotes the $i$-th categorical distribution in the state space. The absorbing states correspond to the $K$ distributions with $p_k = 1$ for one $k$ and $p_i = 0$ for all other $i$, which again have zero variance, i.e. are collapsed distributions. $\qquad\square$

Proposition C.3 is a special case of the argument of Shumailov et al. (2023) that iterative relearning with discrete distributions describes an absorbing Markov chain, which is known to converge to absorbing states with probability 1. Our proof explicitly constructs the underlying absorbing Markov chain for categorical distributions.

**Expected number of steps until model collapse.** Interestingly, with a single sample the transition matrix is the identity matrix and the distribution collapses immediately. In general, more samples means slower collapse. Specifically, the expected steps until model collapse corresponds to the expected steps to reach an absorbing state and can be computed by the fundamental matrix $\sum_{t=0}^{\infty} Q^t$, where $Q$ is the submatrix of the transition matrix $P$ corresponding to the transient states.

Notably, the submatrix $Q$ is a strictly substochastic matrix, i.e. the sum of the entries in each row is strictly less than one (since it does not contain the non-zero probability of transitioning to absorbing states). We can bound the eigenvalues of $Q$ using the Gershgorin circle theorem (Geršgorin, 1931), which states that every eigenvalue of a square matrix $M$ lies within a closed disk centered at $M_{ii}$ with radius $R_i$, where $M_{ii}$ is the diagonal element of $M$ and $R_i$ is the sum of the absolute values of the off-diagonal elements of row $i$, $R_i = \sum_{j \neq i} |M_{ij}|$. In our case, since $Q$ is substochastic, the absolute eigenvalues of $Q$ are strictly less than one. This allows us to apply the geometric series of matrices and compute the

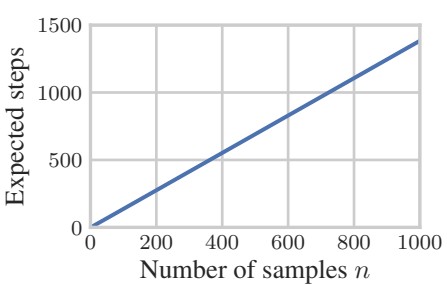

Figure 31: Expected number of steps until model collapse for Bernoulli distributions.

fundamental matrix as $\sum_{t=0}^{\infty} Q^t = (I - Q)^{-1}$. The expected number of steps until model collapse can be computed by solving the linear system $(I - Q)\mathbf{t} = \mathbf{1}$. Overall, starting in transient state $i$, the expected number of steps until model collapse is given by $\mathbf{t_i}$. In Figure 31 we show that the expected steps $\mathbf{t}_i$ grows linearly with the number of samples $n$.

**From model collapse to machine unlearning for categorical distributions.**

**Lemma 1:** *For any categorical distribution $\pi_0$, iteratively relearning $\pi_t$ on target data $\mathcal{D}_\mathcal{C}$ augmented with data generated from its own distribution $\{x_i \mid x_i \sim \pi_t\}_{i=1}^n$ causes information loss for all other (non-target) categories $i$: $\pi_t(i) \xrightarrow{t \to \infty} 0$.*

*Proof.* Because of the fixed retain set, probabilities for retain categories remain non-zero, while probabilities for all other categories can become zero if no samples from these categories are generated during the iterative relearning process. Once the probability of a category becomes zero, it cannot be recovered anymore, since the iterative relearning process only generates samples from the current distribution $\pi_t$ and will not generate samples from categories that have zero probability. This process can be described once again using an absorbing Markov chain, where the absorbing states correspond to the distributions with zero probability for all categories except the retain categories. □

**Beyond categorical distributions.** We empirically demonstrate partial collapse in finite samples for distributions described by Gaussian mixture models (GMMs) for 1- and 2-dimensional data. Specifically, we sample two datasets from two isotropic Gaussians, one retain and one forget set. We then fit a GMM with two Gaussians on the joint dataset to obtain a starting distribution $p_0$. We then iteratively relearn the GMMs either (1) on datapoints sampled from the model's own distribution only, or (2) on retain data augmented with datapoints sampled from the model's own distribution. Figure 32 and Figure 33 show that iterative relearning on self-generated data leads to information loss – either the distribution collapses to zero variance or the variance diverges. In contrast, Figure 34 and Figure 35 show that iterative relearning on retain points augmented with self-generated data leads to partial collapse, i.e. the probability mass of the forget distribution is redistributed to the retain distribution. This process stabilizes and does not collapse. This is consistent with the observation for categorical distributions in Figure 2 (collapse) and Figure 30 (partial collapse/unlearning).

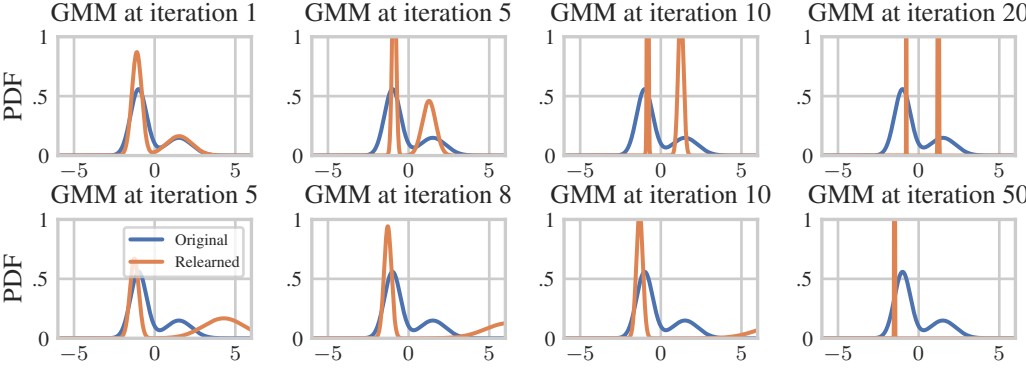

Figure 32: Model collapse during iterative relearning of 1D-GMMs without retain set. Variance of each individual Gaussian either converges to 0 (top row) or diverges to $\infty$ (bottom row) in finite steps.

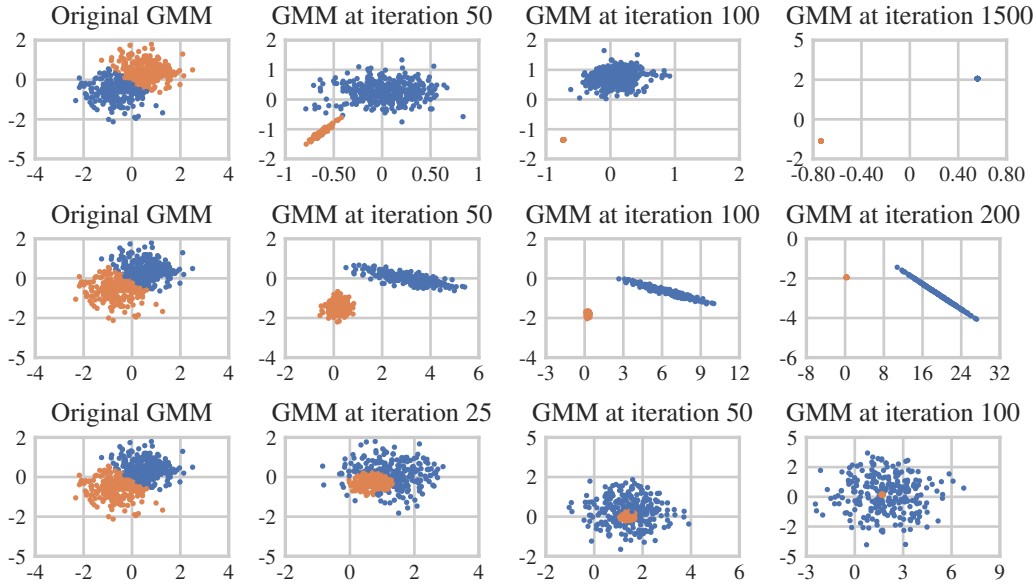

Figure 33: Model collapse during iterative relearning of 2D-GMMs without retain set. Variance of each individual Gaussian either vanishes or diverges (in finite steps).

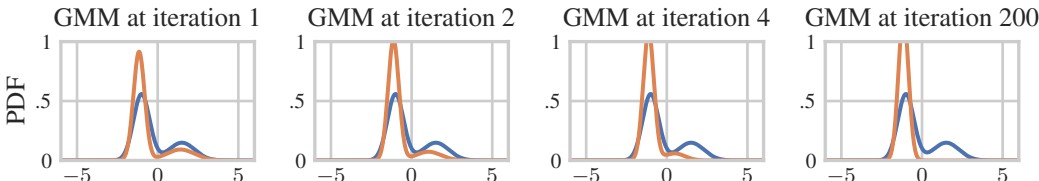

Figure 34: Partial model collapse unlearning for 1D-GMMs: When augmenting retain data with self-generated data, the probability mass of the forget distribution is redistributed to the retain distribution. The iterative relearning process stabilizes and does not collapse.

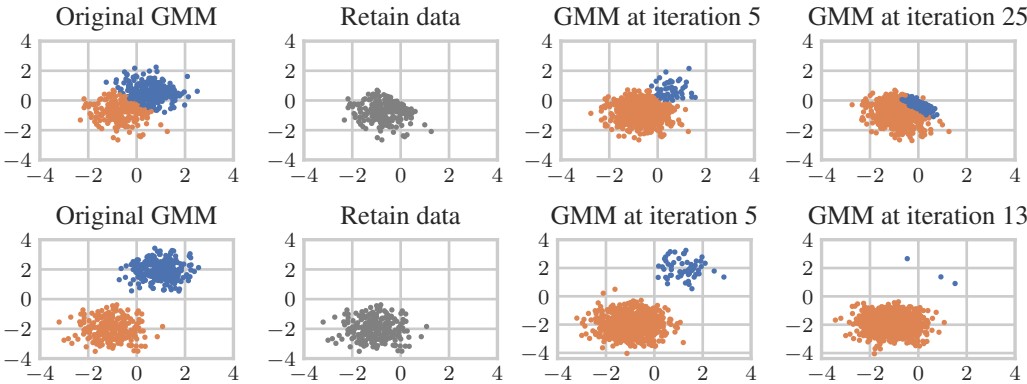

Figure 35: Partial model collapse unlearning for 2D-GMMs: When augmenting retain data with self-generated data, the probability mass of the forget distribution is redistributed to the retain distribution. The iterative relearning process stabilizes and does not collapse. Note that singularities in the EM-algorithm may occur during this iterative process (bottom row).

## D  MACHINE UNLEARNING VIA RELEARNING ON SELF-GENERATED DATA

The approach we describe in Section 4 is specific for Q&A tasks, but we can generalize it into unlearning for arbitrary tasks as well: Let $p_0$ denote the PDF (PMF) of any distribution over a set $\mathcal{X} \subseteq \mathbb{R}^d$. Starting from an initial distribution, the objective is to obtain a model that fits a target distribution $p_r$, while erasing the influence of a forget distribution $p_f$. Given a target distribution $p_r$ over $\mathcal{X}$ that we do not want to unlearn, we propose machine unlearning via iterative relearning as:

---
**Partial Model Collapse Machine Unlearning**

$$p_{t+1} = \arg\min_{p \in \mathcal{P}(\mathcal{X})} \frac{\alpha}{1+\alpha} \mathbb{E}_{x \sim p_r}[-\log p(x)] + \frac{1}{1+\alpha} \mathbb{E}_{x \sim p_t}[-\log p(x)] \tag{3}$$

---

where $\mathcal{P}$ is the set of densities over $\mathcal{X}$, and $\alpha \in [0, \infty)$. Intuitively, Equation 3 describes an iterative process where the next distribution minimizes the convex combination of the expected negative log-likelihood (NLL) under a retain distribution and the expected NLL under the current distribution $p_t$. Notably, this iterative process converges to the retain distribution (Proof in Appendix D):

> **Theorem 2:** *Assuming no statistical approximation errors, $p_t$ converges exponentially with rate $\frac{1}{1+\alpha}$ to the target distribution $p_r$ for any initial distribution $p_0$, $\lim_{t \to \infty} p_t(x) = p_r(x)$.*

Here, larger $\alpha$ yields faster convergence to the retain distribution $p_r$. Notably, we do not require any unlearning target, i.e. this method is independent of any forget distribution. In particular, Theorem 2 implies that for any forget distribution $p_f$ over $\mathcal{X}$ the KL-divergence between $p_t$ and $p_f$ converges to the KL-divergence between retain and forget distribution: $D_{\mathrm{KL}}(p_t \| p_f) \xrightarrow{t \to \infty} D_{\mathrm{KL}}(p_r \| p_f)$.

*Proof.* Due to assumption 1, we can express the PDF of the iterative relearning scheme as follows:

$$p_t(x) = \frac{\lambda}{1+\lambda} p_r(x) + \frac{1}{1+\lambda} p_{t-1}(x)$$

since the assumption ensures $q = \arg\max_{p \in \mathcal{P}} \mathbb{E}_{x \sim q}[\log p(x)]$. Note this is a recursion equation for which we can derive a closed-form:

$$
\begin{aligned}
p_t(x) &= \frac{\lambda}{1+\lambda} p_r(x) + \frac{1}{1+\lambda} p_{t-1}(x) \\
&= \frac{\lambda}{1+\lambda} p_r(x) + \frac{1}{1+\lambda} \left( \frac{\lambda}{1+\lambda} p_r(x) + \frac{1}{1+\lambda} p_{t-2}(x) \right) \\
&\vdots \\
&\overset{(1)}{=} \frac{\lambda}{1+\lambda} \sum_{i=0}^{t-1} \left( \frac{1}{1+\lambda} \right)^i p_r(x) + \left( \frac{1}{1+\lambda} \right)^t p(x) \\
&\overset{(2)}{=} \frac{\lambda}{1+\lambda} \frac{1 - \left( \frac{1}{1+\lambda} \right)^t}{1 - \frac{1}{1+\lambda}} p_r(x) + \left( \frac{1}{1+\lambda} \right)^t p(x) \\
&\overset{(3)}{=} \left[ 1 - \left( \frac{1}{1+\lambda} \right)^t \right] p_r(x) + \left( \frac{1}{1+\lambda} \right)^t p(x)
\end{aligned}
$$

where in (1) we insert the initial distribution $p_0(x) = p(x)$ after unrolling all $t$ iterations, in (2) we use the geometric sum using $\lambda > 0$ and thus $\frac{1}{1+\lambda} \in (0, 1)$, and in (3) we just simplify the expression $\frac{1}{1 - \frac{1}{1+\lambda}} = \frac{1+\lambda}{\lambda}$.

Thus we have derived a closed-form of $p_t(x)$:

$$p_t(x) = \left[ 1 - \left( \frac{1}{1+\lambda} \right)^t \right] p_r(x) + \left( \frac{1}{1+\lambda} \right)^t p(x)$$

Using this closed-form of $p_t(x)$ we directly obtain the convergence of $p_t(x)$ for $t \to \infty$:

$$p_\infty \triangleq \lim_{t \to \infty} p_t(x) = p_r(x)$$

since due to $\lambda > 0$ we have $\frac{1}{1+\lambda} \in (0,1)$ and thus $\left(\frac{1}{1+\lambda}\right)^t \xrightarrow{t \to \infty} 0$.

Consequently we further have:

$$D_{\mathrm{KL}}(p_\infty || p_r) = \mathbb{E}_{p_\infty}\left[\log \frac{p_\infty(x)}{p_r(x)}\right] = \mathbb{E}_{p_r}\left[\log \frac{p_r(x)}{p_r(x)}\right] = \mathbb{E}_{p_r}[\log 1] = 0$$

and

$$D_{\mathrm{KL}}(p_\infty || p_f) = \mathbb{E}_{p_\infty}\left[\log \frac{p_\infty(x)}{p_f(x)}\right] = \mathbb{E}_{p_r}\left[\log \frac{p_r(x)}{p_f(x)}\right] = D_{\mathrm{KL}}(p_r || p_f)$$

and specifically for mutually exclusive support of $p_r$ and $p_f$ we have:

$$D_{\mathrm{KL}}(p_\infty || p_f) = D_{\mathrm{KL}}(p_r || p_f) = \infty$$

$\square$

Finally, we prove the theorem about the expected reward convergence and vanishing variance for the iterative relearning as described by Equation 2:

$$p_{t+1} = \arg\max_{p \in \mathcal{P}} \lambda \mathbb{E}_{(q,x) \sim p_r}[\log p(x|q)] + \mathbb{E}_{\substack{q \sim p_f \\ x_1, \ldots, x_n \sim p_t(x|q) \\ \hat{x} \sim \mathcal{BT}_\tau(x_1, \ldots, x_n)}}[\log p(\hat{x}|q)]$$

**Theorem 1:** *Let $p_t$ be the distribution described by Equation 2 and assume non-zero probability mass on the maximum reward $\Pr_{x \sim p_0(x|q)}[r(x) = r^*] > 0$ for forget queries $q \in supp(p_f)$. In the absence of statistical and function approximation errors, the expected reward converges to the maximum reward and its variance vanishes for any forget query $q \in supp(p_f)$:*

$$\mathbb{E}_{x \sim p_t(x|q)}\left[e^{r(x)}\right] \xrightarrow{t \to \infty} e^{r^*} \qquad \mathrm{Var}_{x \sim p_t(x|q)}\left[e^{r(x)}\right] \xrightarrow{t \to \infty} 0.$$

*Proof.* We consider the following iterative optimization problem (Equation 2):

$$p_{t+1} = \arg\max_{p \in \mathcal{P}} \lambda \mathbb{E}_{(q,x) \sim p_r}[\log p(x|q)] + \mathbb{E}_{\substack{q \sim p_f \\ x_1, \ldots, x_n \sim p_t(x|q) \\ \hat{x} \sim \mathcal{BT}(x_1, \ldots, x_n)}}[\log p(\hat{x}|q)]$$

Assuming no statistical approximation errors, we know that $\arg\max_{p \in \mathcal{P}} \mathbb{E}_{x \sim q}[\log p(x)] = q$. In the case of conditional distributions we have $\arg\max_{p \in \mathcal{P}} \mathbb{E}_{(q,x) \sim p_r}[\log p(x|q)] = p^*$ with $p^*(x|q) = p_r(x|q)$. Since we assume that the supports of $p_r$ and $p_f$ are disjoint, the optimization problem amounts to two independent problems and the density of the optimal distribution $p_{t+1}^*$ matches each conditional distribution independently:

$$p_{t+1}^*(x|q) = \begin{cases} p_r(x|q) & \text{if } q \in supp(p_r) \\ \hat{p}_{t+1}(x|q) & \text{if } q \in supp(p_f) \end{cases}$$

where $\hat{p}(x|q)$ is the distribution that maximizes the second term in Equation 2 for $q \in supp(p_f)$:

$$\hat{p}_{t+1}(x|q) = \arg\max_{p \in \mathcal{P}} \mathbb{E}_{\substack{q \sim p_f \\ x_1, \ldots, x_n \sim \hat{p}_t(x|q) \\ \hat{x} \sim \mathcal{BT}(x_1, \ldots, x_n)}}[\log p(\hat{x}|q)]$$

Assuming again no statistical approximation errors, one can show that the density of the distribution $\hat{p}_{t+1}(x|q)$ assumes a closed-form (proof in (Ferbach et al., 2024) – proof of Lemma 2.1):

$$\hat{p}_{t+1}(x|q) = \hat{p}_t(x|q) \cdot H_{\hat{p}_t}^n(x|q)$$

with

$$H_{\hat{p}_t}^n(x|q) = \mathbb{E}_{x_1, \ldots, x_{n-1} \sim \hat{p}_t(x|q)}\left[\frac{n e^{r(x)}}{e^{r(x)} + \sum_{i=1}^{n-1} e^{r(x_i)}}\right].$$

Moreover, since we assume the reward is bounded, Assumption 2.1 in (Ferbach et al., 2024) holds and consequently the statement about reward convergence and vanishing variance follows directly from Lemma 2.2 in (Ferbach et al., 2024). $\square$

