# OpenReview forum: "Model Collapse Is Not a Bug but a Feature in Machine Unlearning for LLMs"
_ICLR.cc/2026/Conference — ICLR 2026 Poster_

### Official Review · Reviewer_G3SG · 2025-10-28

**Soundness:** 3
**Presentation:** 2
**Contribution:** 2
**Rating:** 4
**Confidence:** 4

**Summary:**

The paper proposes Partial Model Collapse (PMC), a new machine unlearning method for large language models. The key idea is to deliberately induce controlled model collapse by fine-tuning a model only on its own generated outputs for questions whose answers must be forgotten. Unlike existing empirical unlearning methods, PMC does not require access to or optimization directly over the private “forget” answers, which the authors claim reduces leakage risk and preserves privacy. The paper includes a theoretical formulation showing convergence toward a collapsed output distribution and presents empirical results on the TOFU benchmark indicating improved unlearning-utility trade-offs and reduced side effects.

**Strengths:**

The paper challenges the dominant paradigm of unlearning systems that explicitly suppress or optimize against ground truth sensitive answers, proposing a more privacy-aligned direction: remove private knowledge without touching the private data itself. This direction is well-motivated in regulatory contexts such as GDPR. The high-level design is simple and connects strongly with recent discussions about distribution collapse in synthetic self-training. Empirical results show improvements on several relevant axes including utility retention, robustness under sampling attacks, and reduced unintended suppression of semantically similar tokens. These highlight that the authors care not only about benchmark metrics but also about subtle security implications.

**Weaknesses:**

There are notable limitations in the current form. First, the theoretical results are built on highly abstract distributions and reward convergence assumptions, and the paper does not convincingly bridge these assumptions with actual LLM behavior. Second, all experiments are on a synthetic dataset (TOFU), so claims of “privacy-preserving” remain hypothetical. Third, some baselines are insufficiently understood or tuned to ensure fairness: certain hyperparameter choices appear inconsistent with prior literature. Fourth, the collapse mechanism could potentially degrade knowledge in areas semantically close to the forget domain, yet the evaluation of utility is relatively narrow, relying mostly on ROUGE. Last, computational overhead implications are minimized rather than rigorously quantified.

**Questions:**

It would help to clarify how PMC avoids “over-forgetting” in more complex semantic spaces. For example, if the model self-generates answers that gradually omit related but still essential information, how do you detect and prevent cascading collapse? Better visualization of token-level drift could strengthen this point.

The paper acknowledges that computational cost of repeated sampling is a bottleneck, but offers no concrete profiling or mitigation strategies. Could a smaller proxy model or distillation-based filtering accelerate collapse while preserving safety? Please include runtime vs unlearning efficacy trade-offs.

On the theoretical side, Theorem 2 depends on no statistical or function approximation error. LLMs always operate under approximation, so the guarantee is weakened. Could you provide empirical convergence plots matching the theorem’s predicted reward evolution?
The TOFU benchmark, while convenient, lacks realistic attack surfaces and nuanced sensitive information patterns. Any evidence on real-world factual unlearning tasks would dramatically improve credibility.

Hyperparameter search for baselines might be unintentionally restrictive. For example, NPO often shows higher stability when λ is dynamically adjusted. Please clarify search ranges and why they fairly reflect best-practice settings.

For privacy claims, you state that PMC avoids using sensitive data. However, model self-generations may already encode the sensitive data. Under what conditions does PMC not reinforce hidden memorization before collapse? More discussion of this risk is needed.

---

> ### Author Response · Authors · 2025-11-20
>
> Thank you for your review!
>
> **Regarding over-forgetting.** Our analysis directly addresses this point. In Section 5.2, we quantify token-level drift and show that PMC does not introduce a bias against tokens associated with the unlearning set (Figure 4 a). In contrast, prior approaches show clear token-level bias indicating over-forgetting. Our empirical findings clearly show that PMC is substantially more robust to such failure modes, and in particular, does not trigger cascading collapse. In response to your comment, we will highlight these observations more explicitly in the camera-ready version to clarify the advantages of PMC in avoiding over-forgetting.
>
> **Regarding connecting theory and practice.** Thank you for pointing out potential to further connect theory and practice. In response, we provide additional empirical evidence that the observed reward quickly converges to the maximum reward within the first two epochs in Appendix C.2. This empirically validates our theoretical result in Theorem 2. Beyond that, please note that our main theoretical contribution is to establish a formal connection between the model collapse and machine unlearning fields, enabling transfer of insights and guarantees. We believe establishing this connection is non-trivial and meaningful. The assumptions we adopt also align with those commonly used in the model-collapse literature, where developing results under more realistic assumptions is a highly active research area. We believe our insights are highly valuable to the community and represent an important step toward a deeper understanding of machine unlearning.
>
> **Regarding an extended costs analysis.** Thank you for helping us to improve the empirical analysis. In response, we provide a careful and detailed cost analysis in Appendix C.3, comparing PMC against baselines under runtime vs. unlearning and utility trade-offs. While we acknowledge that PMC has a slightly higher computational cost during the initial collapse process, we believe the overall runtime remains competitive and in particular practical for real-world applications. While the core contribution of our paper is to establish the connection between model collapse and machine unlearning, we believe optimizing PMC in terms of computational cost is a promising direction for future research.
>
> **Regarding additional evidence.** We appreciate your request for additional empirical evidence. In response, we extended the evaluation of our approach to a subset of MUSE as described in Appendix C.1., where our collapse-based unlearning method again outperforms the baselines. We will further extend this analysis for the camera-ready version. That said, we believe our careful and thorough analysis on TOFU is of high value to the community, in particular our discussion of the failure modes that we uncover in previous approaches (as also highlighted by other reviewers).
>
> **Regarding hyperparameter ranges.** Please note that we carefully document all hyperparameter ranges of our grid search in Appendix B.2, which uses (and partially exceeds) the ranges used e.g. in the SimNPO paper. Importantly, we do not merely tune hyperparameters; instead, we explore the entire Pareto-front under 100 different hyperparameter configurations. This provides a strictly stronger basis for comparison than the single-model results reported in prior work. Moreover, we repeat all experiments five times to account for randomness. While baselines might benefit from advanced tuning beyond the commonly described setups in the literature, the same holds for our approach. Overall, we believe our experimental setup is fair, reproducible, and in particular adequate for scope of our work.
>
> **Regarding sensitive data.** Thank you for your comment, we acknowledge that self-generations may encode sensitive information. The key advantage of our approach is, however, that PMC ensures the model quickly diverges from such data, both from a theoretical perspective and in our empirical study. After the initial divergence phase, PMC no longer exposes the model to sensitive data during optimization. In contrast, all previous GA-based methods repeatedly fine-tune on the ground truth throughout the entire unlearning process, thereby amplifying such signals as we discuss in Section 5.2. In response to your comment, we will clarify this more explicitly in the camera-ready version. Beyond that, we would also like to emphasize that our setting is practically relevant: For example, one might simply not know which training data exactly contributed to a particular model output that requires unlearning, or the model might even infer answers to critical questions without a particular ground truth. Overall, we believe our approach is an important step toward more privacy-preserving unlearning that avoids reinforcing sensitive information.
>
> We hope that we could address all your comments to your satisfaction. Please let us know if you have any additional questions.

---

### Official Review · Reviewer_PRv6 · 2025-10-29

**Soundness:** 3
**Presentation:** 3
**Contribution:** 3
**Rating:** 4
**Confidence:** 3

**Summary:**

This paper introduces Partial Model Collapse (PMC), a novel paradigm for machine unlearning in large language models (LLMs). Unlike existing methods that rely on direct optimization against sensitive unlearning targets, PMC leverages the natural information loss inherent in model collapse—the degradation observed when models are repeatedly fine-tuned on self-generated data. The core idea is to intentionally trigger partial collapse in response to sensitive queries, allowing the model to “forget” without reusing private data.

**Strengths:**

- The paper reinterprets model collapse, typically seen as undesirable, into a constructive mechanism for unlearning—an elegant and theoretically grounded insight.

- Clear mathematical derivation of convergence properties (Lemma 1, Theorem 2) and ablation studies validating the hyperparameter behavior.

- The paper is clearly structured, visually intuitive, and well-written.

**Weaknesses:**

- While conceptually elegant, PMC requires multiple sampling and fine-tuning iterations, potentially increasing computational cost. Could the authors quantify runtime and explore lightweight approximations?

- The method’s performance hinges on the choice of reward function r(x). How robust is PMC to alternative reward definitions, and can it generalize beyond ROUGE-based divergence metrics?

- The theoretical analysis assumes idealized convergence. How does PMC behave when unlearning large sets simultaneously or when errors accumulate over many collapse steps?

**Questions:**

Refer to the Weakness above.

---

> ### Author Response · Authors · 2025-11-20
>
> Thank you for your review!
>
> **Regarding an extended costs analysis.** In response to your comment, we provide a careful and detailed cost analysis in Appendix C.3, comparing PMC against baselines under runtime vs. unlearning and utility trade-offs. While we acknowledge that PMC has a slightly higher computational cost due to the initial collapse process in the first two epochs, we believe that the overall runtime remains competitive and in particular practical for real-world applications.
>
> **Regarding empirical convergence results.** Thank you for pointing out potential to further bridge the gap between theory and practice. In response to your comment, we provide additional empirical evidence that the observed reward quickly converges to the maximum reward within the first two epochs as shown in Figure 9 in Appendix C.2. This additional experiment empirically validates our theoretical result in Theorem 2. Beyond that, please note that our main theoretical contribution is to establish a formal connection between the model collapse and machine unlearning literatures, enabling the transfer of insights and guarantees from one field to the other. We believe establishing this connection is non-trivial and meaningful. The assumptions we adopt also align with those commonly used in the model-collapse literature, where developing results under more realistic assumptions is a highly active research area. We believe that our core insights are highly valuable to the unlearning community and represent an important step toward a deeper theoretical understanding of machine unlearning.
>
> **Regarding faster approximations.** Thank you for your question. In response, we implemented a faster approximation making use of the previous observation that the empirical reward converges quickly. The main idea is to only select samples if their reward is larger than that of the sample with the best reward observed so far, and to stop sampling once the reward has converged to the maximum reward. This can speed up convergence, significantly reduces the number of samples required during unlearning, and achieves comparable unlearning and utility as the vanilla approach (see Appendix C.3). Beyond that, we believe PMC can be further optimized in terms of computational cost by future research, e.g., by developing more efficient sampling strategies, utilizing compression techniques, or making use of parallelization.
>
> **Regarding alternative reward functions.** Please note that we chose ROUGE-based rewards in our main experiments primarily for their efficiency and interpretability. In practice, we find PMC-unlearning to be quite robust to alternative reward formulations. For example, we experimented with extending ROUGE scores by adding a penalty for overly short answers (encouraging longer outputs after unlearning) and observed comparable performance. Beyond ROUGE, we also explored reward functions based on self-BLEU scores between samples, which we find to yield effective unlearning as well. We ultimately favored ROUGE-scores in our main experiments for the sake of simplicity. In response to your comment, we will include an additional ablation study for the reward function in the camera-ready version.
>
> We hope that we could address all your questions to your satisfaction. Please let us know if you have any additional comments or questions.

---

> > ### Comment · Reviewer_PRv6 · 2025-11-27
> >
> > Thank you for your reply, it solved my question, and I have improved my score.

---

### Official Review · Reviewer_ugX6 · 2025-10-31

**Soundness:** 3
**Presentation:** 3
**Contribution:** 3
**Rating:** 6
**Confidence:** 4

**Summary:**

The paper introduces Partial Model Collapse (PMC), a learning-to-forget procedure that reframes machine unlearning as iterative relearning on self-generated samples. It presents a practical algorithm for LLMs that avoids using ground-truth forget answers by sampling candidate responses and updating on a preference-selected sample while jointly training on retain data. Theoretically, the paper proves convergence properties of the iterative process under idealized assumptions, and motivates the approach via categorical-distribution warm-ups. Empirically, PMC expands the Pareto front between unlearn quality and utility.

**Strengths:**

- The paper offers a interesting perspective: leveraging model collapse via iterative relearning on self-generated data as a mechanism for unlearning, with a clear derivation from categorical settings and a principled LLM instantiation.

- The theoretical section establishes convergence of the reward and vanishing variance for the iterative update under stated assumptions, providing a clear link between the objective and unlearning behavior.

- The algorithmic procedure is explicit and the narrative emphasizes why avoiding ground-truth forget answers can prevent unintended reinforcement.

- PMC expands the utility–unlearn-quality Pareto front for backbone model, indicating practical promise.

**Weaknesses:**

- Evaluation scope is narrow. Experiments focus on a single unlearning benchmark (TOFU), two LLMs (Phi-1.5, Llama-3.2-3B-Instruct), limiting generality. Additional datasets and tasks would bolster claims.

- Computational cost. The method depends on sampling from the model distribution and the paper acknowledges overhead for LLMs. A clearer cost–benefit analysis or experimental comparisons versus baselines would enhance soundness.

- Assumptions behind the theory are strong. Theoretical arguments rely on idealized settings, e.g., no approximation error, rendering parts of the proofs rather trivial.

**Questions:**

1. How does PMC perform under prompt-injection attack?

2. Is PMC able to achieve superior efficiency in the trade-off between computational resources and performance benefits relative to the baseline models?

---

> ### Author Response · Authors · 2025-11-20
>
> Thank you for your review!
>
> **Regarding additional evaluations.** We appreciate your request for additional empirical evidence. In response to your comment, we extend the experimental analysis to an additional dataset (MUSE) and model (Llama-2-7b-hf), as described in detail in Appendix C.1 of the updated manuscript. In this additional evaluation our collapse-based unlearning method again outperforms the baselines. We will extend this empirical analysis for the camera-ready version and also add results for Gemma3-12B (please consider that a complete hyperparameter search for all baselines for this model is not possible within the scope of the rebuttal). That said, we believe our careful and thorough empirical analysis on TOFU is of high value to the community, in particular our discussion of the failure modes that we uncover in previous approaches (as also highlighted by other reviewers).
>
> **Regarding an extended cost analysis (W2) and efficiency trade-offs (Q2).** Thank you for your comment and question. In response, we provide a careful and detailed cost analysis in Appendix C.3, comparing PMC against baselines under runtime vs. unlearning and utility trade-offs. While we acknowledge that PMC has a slightly higher computational cost due to the initial collapse process in the first two epochs, we believe that the overall runtime remains competitive and in particular practical for real-world applications. We also believe that PMC can be further optimized in terms of computational cost by future research, e.g., by developing more efficient sampling strategies, utilizing compression techniques, or making use of parallelization.
>
> **Regarding the theoretical assumptions.** Please note that our main theoretical contribution is to establish a formal connection between the model collapse and machine unlearning literatures, enabling the transfer of insights and guarantees from one field to the other. We believe establishing this connection is non-trivial and meaningful. The assumptions we adopt also align with those commonly used in the model collapse literature, where developing results under more realistic assumptions is a highly active research area. Importantly, future convergence results obtained in model collapse can transfer to unlearning, which we believe makes our theoretical insights particularly valuable.
>
> **Regarding prompt-injection attacks.** Thank you for your question. Our empirical analysis in Section 5.3 shows that PMC is substantially more robust to prefilling attacks, a form of prompt-injection that has been shown to be considerably strong compared to adversarial attacks such as GCG (see e.g. [1]). While we consider additional threat models and attacks as a promising avenue for future research, we believe our robustness analysis provides valuable insights for the unlearning community.
>
> We hope that we could address all your questions to your satisfaction. Please let us know if you have any additional comments or questions.
>
>
> [1] Jailbreaking Leading Safety-Aligned LLMs with Simple Adaptive Attacks. ICLR 2025.

---

### Official Review · Reviewer_Jbgi · 2025-11-01

**Soundness:** 3
**Presentation:** 4
**Contribution:** 3
**Rating:** 6
**Confidence:** 3

**Summary:**

The study introduced PMC, Partial Model Collapseas, as a effective unlearning method for LLMs. PMC is to let a model relearn on its own generated outputs on targeted queries. This self-training gradually collapses the model’s confidence on unwanted responses while maintaining general utility.

**Strengths:**

Overall, the study offers a novel unlearning approach based on model collapse, which is often viewed as a defect. The proposed PMC method is original in both formulation and intuition, achieving unlearning without relying on the sensitive information that needs to be removed. The preference-guided self-training mechanism is also an interesting idea.

The technical quality of this paper is strong, with a solid theoretical foundation and convincing empirical validation on the benchmark. The comparisons with prior methods, e.g. GA, GD, NPO, and SimNPO, clearly demonstrate the effectiveness of PMC. The ablation studies on temperature, sample count, and weighting are also thorough and well executed.

The paper is well written and easy to follow. The overall presentation is clear and well organized.

**Weaknesses:**

The experiments rely solely on the TOFU dataset, which is somewhat limiting. It would be beneficial to validate the performance on additional benchmarks such as MUSE, WMDP, or others to strengthen the empirical evidence.

PMC is compared against GA, GD, NPO, and SimNPO, but not against several recent unlearning methods, such as SCRUB, DPO, or Negative Preference Fine-Tuning. Including these comparisons would provide a more comprehensive evaluation of the proposed approach.

It would also be valuable to discuss the potential applicability and performance of PMC in other domains, such as image or tabular data, to better understand its generality.

Moreover, the discussion and analysis of computational cost could be expanded, including runtime, resource usage, and scalability with model size and data volume.

**Questions:**

L132, "In this work, we study empirical machine unlearning for generative models..." Should that be just LLM?

" Note that we do not require access to the ground truth answers for the forget questions, and we assume disjoint support of pf(q) and the marginal distribution pr(q), i.e. we either want to unlearn the response to a question or not." What if there are overlaps between responses to be removed and not to be removed?

L840, minor latex issue, ’Yes’ -> `Yes’

---

> ### Author Response · Authors · 2025-11-20
>
> Thank you for your review!
>
> **Regarding additional evaluations.** We appreciate your request for additional empirical evidence. In response to your comment, we extend the empirical evaluation of our approach to a subset of MUSE, as described in detail in Appendix C.1 of the updated manuscript. In this experiment, our collapse-based unlearning method again outperforms the baselines. We will further extend the empirical analysis for the camera-ready version. That said, we believe that our careful and thorough empirical analysis on TOFU is of high value to the community, in particular our discussion of the failure modes that we uncover in previous approaches (as also highlighted by other reviewers).
>
> **Regarding other unlearning methods.** Our evaluation includes very recent state-of-the-art LLM unlearning approaches that are directly applicable to the natural language domain. This includes, in particular, the NPO baseline that is directly derived from DPO and performs negative preference optimization. In this context, DPO is conceptually very similar to our IDK baseline, which comes with significant drawbacks as outlined in Section 5.3. Nevertheless, we appreciate the comment, agree that additional baselines can be helpful, and will include DPO in the camera-ready version for completeness. Moreover, please note that SCRUB is an unlearning method for image classification and not directly applicable to LLM unlearning so far.
>
> **Regarding the applicability to other domains.** Thank you for your suggestion to further discuss the applicability of PMC to other domains. We believe that our central idea of leveraging model collapse for machine unlearning generalizes beyond language, in particular since model collapse has been observed not only in LLMs but also in image-generation models.  While we agree that adapting our approach to additional domains is a promising avenue for future research, we would like to emphasize that our paper is addressing a particularly pressing need for unlearning in LLMs. We appreciate this comment and will extend the discussion in camera-ready version of the paper.
>
> **Regarding an extended costs analysis.** In response to your comment, we provide a careful and detailed cost analysis in Appendix C.3, comparing PMC against baselines under runtime vs. unlearning and utility trade-offs. While we acknowledge that PMC has a slightly higher computational cost due to the initial collapse process in the first two epochs, we believe that the overall runtime remains competitive and in particular practical for real-world applications. We also believe that PMC can be further optimized in terms of computational cost by future research, e.g., by developing more efficient sampling strategies, utilizing compression techniques, or making use of parallelization.
>
> **Regarding the overlap assumption.** Please note that our assumption only concerns the questions, not the responses: we assume that forget and retain questions are separate, meaning that each question is either part of the retain or the forget set. This assumption is natural for Q&A unlearning, and  importantly, potential overlaps in the responses to questions are not an issue and therefore also not part of the assumption. As we demonstrate in Appendix C.1, our main insight of using model collapse for machine unlearning also generalizes beyond Q&A tasks to datasets like MUSE, highlighting the broader applicability of our approach across different unlearning settings.
>
> Thank you, we fixed the minor points in L132 and L840 in the updated manuscript.
>
> We hope that we could address all your questions to your satisfaction. Please let us know if you have any additional comments or questions.

---

> > ### Comment · Reviewer_Jbgi · 2025-11-28
> >
> > Thank the author for the further explanation and additional experiments. My comments have been addressed to some extent. I incline to maintain my score, and encourage the author to extend the work further.

---

> > > ### Author Response · Authors · 2025-11-28
> > >
> > > Thank you for your follow-up. We appreciate the positive assessment of our work and are glad that our additional explanations and experiments addressed your comments. If there are remaining concerns, we would be thankful for any further clarification. We would like to highlight that the core contribution of our paper lies in the new perspective and the first combination of these two research directions. The additional experiments presented in the rebuttal further support our main findings, and we believe that any further extensions would not change the core message of our work. We also believe that our novel perspective is highly valuable for both communities (unlearning and model collapse) and opens up promising avenues for future research.

---

### Author Response · Authors · 2025-11-20
**Global response**

We would like to thank all reviewers for their valuable feedback and for acknowledging the novelty of our approach for machine unlearning! We changed our initial submission in response to their comments as follows:

- Additional computational cost analysis in Appendix C.3 in response to Reviewers Jbgi, ugX6, PRv6 and G3SG.
- Additional evaluations in Appendix C.1 providing empirical evidence on an additional dataset (MUSE) in response to Reviewers Jbgi, ugX6 and G3SG.
- Additional demonstration of empirical convergence in Appendix C.2 in response to Reviewers PRv6 and G3SG.
- Fixed two minor typos in response to Reviewer Jbgi.

---

### Author Response · Authors · 2025-12-02
**Concluding response**

We would like to thank all reviewers again for their valuable feedback and, in general, positive assessment of our work. In the following we would like to wrap up the discussion.

All reviewers acknowledge the novelty of our approach for machine unleraning.
In response to their comments we have provided (1) an extended analysis of computational cost, (2) empirical evidence on an additional dataset (beyond Q&A), and (3) empirical evidence of theoretical convergence (as outlined in the global response below). Our additional experiments further support and strengthen our main findings.

While we are preparing few results for the final version (including experiments with an additional, even larger model), we believe any further extension would not change the core message of our work. In particular, we see the various possible future research directions (such as different data modalities) as a core strength of our work.

Overall, we would like to highlight that the central contribution of our paper lies in the first combination of two previously separate research fields, unlearning and model collapse, which results in a new and effective framework for machine unlearning. We believe that our novel perspective is highly valuable for both communities (unlearning and model collapse) and opens up promising avenues for future research.

---

### Meta-Review · Area_Chair_EgGM · 2026-01-06

**Summary:**

This submission makes a novel connection between model collapse and machine unlearning in the context of generative models such as LLMs and question-answering tasks, and proposed a novel unlearning scheme --- Partial Model Collapse or PMC --- that does not require the explicit use of the ground-truth answers for the questions in the forget set. The submission presents a multi-faceted unlearning evaluation on the TOFU dataset, highlighting the strong utility-unlearning tradeoff of PMC compared to existing unlearning baselines, as well as novel failure modes for existing unlearning schemes (which the proposed PMC continues to be robust). All reviewers acknowledge this novelty, and the well-structured and high-quality writing in the submission.

However, all reviewers raised the following concerns:

- The evaluation is limited to a single unlearning dataset TOFU. Thus, while the results demonstrate extremely strong performance of the proposed PMC, it is not clear whether the conditions underlying the motivations behind PMC would hold for unlearning setups other than TOFU.
  - The authors responded to this by providing an additional evaluation on a subset of the MUSE dataset, demonstrating similar strong performance for the proposed PMC. The authors also argue that the thorough multi-faceted evaluation on the TOFU dataset is also of high value in its own accord.
  - In my opinion, this partially addresses the issue though a thorough evaluation on MUSE and WMDP benchmarks would completely address this issue.
- The computational overhead of the proposed PMC requiring multiple rounds of generations for each forget set query is not clear, and an proper evaluation of approximate unlearning requires analysis of the unlearning-utility-runtime tradeoff.
  - The authors responded by providing a more detailed evaluation of the computational costs of PMC relative to the NPO and SimNPO baselines.
  - One interesting aspect of the additional results presented in Appendix C.3 is the sudden dramatic initial drop in the utility of the model for PMC, which the model recovers as the unlearning continues. However, this implies that, within a 10-15 minute unlearning time-frame (for the models considered here), the unlearning quality of the proposed PMC and the baselines SimNPO and NPO are comparable, while the utility of the proposed PMC is significantly worse. This is one aspect of the unlearning dynamics that was not apparent in the initial analysis, and is made clear with this additional evaluations of computational costs. The current analysis also does not show the interplay of the balancing parameter $\lambda$, where a smaller value of $\lambda$ might extend the amount of time required for the utility recovery.
  - If we are allowing additional time, fine-tuning on the retain set (with similar amount of time) is also an important baseline to consider. It is known that fine-tuning only on the retain set can induce catastrophic forgetting on the forget set, thereby leading to empirical unlearning, and extended fine-tuning can improve unlearning efficacy. Thus, it is important to show that, for the same amount of unlearning time, the proposed PMC can significantly outperform unlearning via fine-tuning only on the retain set.
  - Thus, in my opinion, this concern is again only partially addressed as the response itself raised some more questions (for example, in terms of the intermediate time-frame performance of PMC and the consideration of other baselines such as fine-tuning).


Thus, overall, I think this is an interesting submission, making a very intuitive connection between model collapse and machine unlearning, leading to a novel unlearning scheme with extremely strong performance (in a somewhat limited evaluation on only a single benchmark). I am recommending an accept though this can be bumped down if necessary.

**Reviewer Concerns:**

Beyond the reviewer concerns discussed in the **Summary** section, the following are the main additional concerns raised by reviewers:
- One reviewer inquired regarding the robustness of the proposed PMC unlearning scheme to prompt-injection attacks. The authors argued in their response that the prefilling attacks considered in the submission is a form of prompt injection attacks, and PMC shows robust performance against such attacks relative to the baselines considered. I believe this concern is addressed.
- One reviewer inquired regarding the effect of non-ROGUE-L reward function and other alternate reward function to the performance of PMC. The authors responded by claiming that the proposed PMC is agnostic to the reward function (as long as it is aligned with the unlearning objective) and discuss various reward functions that they had tried. However, the revision does not contain actual ablation of the reward functions, so it is not completely clear how robust the performance of PMC is to the choice of the reward function, and why the specific ROUGE-based reward function was considered.
- One reviewer raised a concern regarding the fact that the self-generations, the original ones and the subsequently modified ones (which are used in the reward functions), can already encode the sensitive data. The authors responded to this concern by claiming that, while the answer to be unlearned might still be part of the original self-generations, the PMC unlearning scheme pushes the model quickly away from those answers thanks to the reward function considered, thus arguing that the model is not exposed to the sensitive answers during the optimization. However, this seems inadequate to me since the critical reward function used in the PMC unlearning compares the current self-generations to the original generations, and thus is exposed to the sensitive information throughout the unlearning if the original generations contain sensitive information.
- Finally, all reviewers brought up the gap between the assumptions utilized in the theoretical results and practice, highlighting various assumptions that appear too restrictive. The authors responded to this concern by stating that the assumptions considered are standard in model collapse literature, and the main contribution in the theoretical results is the formal connection between model collapse and machine unlearning. Beyond that, the authors also present new experimental validation of the theoretical results, highlighting that practice matches the predictions of the theory. In my opinion, this concern is adequately addressed.

**Reviewer Scores:**

- Reviewer Jbgi would maintain their score of 6.
- Reviewer PRv6 would increase their initial score of 4 (marginally below acceptance threshold) to possibly a 6 (marginally above) as they say that their concerns have been addressed.
- Reviewer ugX6 scored this submission as a 6 (marginally above the acceptance threshold), and I would think that they would maintain their score or increase it by 1 as their concerns regarding narrow scope of evaluation and analysis of computational cost were only partially addressed.
- Reviewer G3SG scored this submission as a 4 (marginally below the acceptance threshold), and I would think that they would maintain their score or increase it by 1 as their concerns regarding limited evaluation, lack of extended computational cost analysis, and sensitive data in self-generations were only partially addressed.

---

### Decision · Program_Chairs · 2026-01-26

Accept (Poster)